# Long range transport of coarse mineral dust: an evaluation of the Met Office Unified Model against aircraft observations

Natalie G. Ratcliffe[1], Claire L. Ryder[1], Nicolas Bellouin[1], Stephanie Woodward[2], Anthony Jones[2], Ben Johnson[2], Lisa-Maria Wieland[3], Maximilian Dollner[3], Josef Gasteiger[3,*], and Bernadett Weinzierl[3]

[1]Department of Meteorology, University of Reading, Reading, UK
[2]Met Office Hadley Centre, Exeter, UK
[3]University of Vienna, Faculty of Physics, Aerosol Physics and Environmental Physics, Boltzmanngasse 5, 1090 Vienna, Austria
[*]Now at Hamtec Consulting GmbH at EUMETSAT, Darmstadt, Germany

**Correspondence:** Natalie G. Ratcliffe (n.ratcliffe@pgr.reading.ac.uk)

**Abstract.** Coarse mineral dust particles have been observed much further from the Sahara than expected based on theory. They have different impacts to finer particles on the Earth's radiative budget, and carbon and hydrological cycles, though tend to be under-represented in climate models. We use measurements of the full dust size distribution from aircraft campaigns over the Sahara, Canaries, Cape Verde and Caribbean. We assess the observed and modelled dust size distribution over long-range transport at high vertical resolution using the Met Office Unified Model, which represents dust up to 63.2 $\mu$m diameter, greater than most climate models. We show that the model generally replicates the vertical distribution of the total dust mass but transports larger dust particles too low in the atmosphere. Importantly, coarse particles in the model are deposited too quickly, resulting in an underestimation of dust mass that is exacerbated with westwards transport; 20-63 $\mu$m dust mass contribution between 2-3.7 km altitude is underestimated by factors of up to 11 at the Sahara, 140 at the Canaries and 240 at Cape Verde. At the Caribbean, there is negligible modelled contribution of d > 20 $\mu$m particles to total mass, compared to 10% in the observations. This work adds to the growing body of research that demonstrates the need for a process-based evaluation of climate model dust simulations to identify where improvements could be implemented.

## 1 Introduction

Every year, 400-2200 Mt of mineral dust is lifted from the Earth's surface and becomes suspended in the atmosphere (Huneeus et al., 2011). This lofted dust can alter the global radiation budget by directly reflecting and absorbing radiation (Kok et al., 2018), altering cloud properties (Lohmann and Feichter, 2005; Price et al., 2018) and precipitation patterns (Rosenfeld et al., 2008) by activating ice and liquid droplet nucleation. Shao et al. (2011) estimate that 75% of the uplifted dust is deposited on land, providing important nutrients to locations such as the Amazon rainforest (Prospero et al., 2020) as well as altering the surface albedo upon deposition, for example on snow and ice (Dumont et al., 2020; Painter et al., 2007). The remaining dust supplies valuable nutrients to nutrient-poor oceans, potentially resulting in the formation of phytoplankton blooms (Jickells et al., 2005; Dansie et al., 2022). Lofted dust also negatively impacts aviation (Nickovic et al., 2021), energy production,

(Piedra et al., 2018) and human health (Kotsyfakis et al., 2019). Many of these processes are sensitive to particle size.

Coarse (2.5 < d < 10 $\mu$m), super-coarse (10 < d < 62.5 $\mu$m) and giant (d > 62.5 $\mu$m) dust particles (size ranges as reviewed and defined in Adebiyi et al. (2023)) have vastly different impacts on the Earth system than fine (d < 2.5 $\mu$m) particles. The lifetime of dust in the atmosphere decreases exponentially with increasing particle diameter (Kok et al., 2017). Sedimentation varies strongly with particle size and dominantly affects super-coarse and giant particles (Foret et al., 2006). The larger particles are also more susceptible to wet deposition processes as they are efficient in-cloud nucleators of ice (Hoose and Möhler, 2012; Pruppacher and Klett, 2010; Sassen et al., 2003; Adebiyi et al., 2023) and, after undergoing in-cloud chemical processing, liquid water (Nenes et al., 2014; Karydis et al., 2011). Coarser particles are also more likely to be removed by below-cloud scavenging (Jones et al., 2022). Coarser particles decrease the amount of outgoing longwave radiation at the Top-Of-the-Atmosphere (TOA) and increase shortwave absorption in the atmosphere, both of which cause a net warming effect at the TOA (Kok et al., 2018). Larger particles also contain a greater mass of the nutrients which provide vital sustenance for the biosphere (Barkley et al., 2021; Baker et al., 2006; Dansie et al., 2017). Simulating the lifetime and transport range of different sized dust particles in models is therefore key to capturing their various effects and impacts.

Recent field campaigns have revealed that coarse, super-coarse and giant particles are transported further across the Atlantic from the Sahara than expected, given their estimated deposition velocity and amount of time in transit (Ryder et al., 2018; Weinzierl et al., 2017; Ryder et al., 2019; van der Does et al., 2016; Denjean et al., 2016). The processes responsible for this unexpected long range transport are unclear. Additionally, many global climate models (GCMs) do not represent super-coarse or giant particles and fail to represent the mass concentration of coarse particles at any stage of transport (Adebiyi and Kok, 2020; O'Sullivan et al., 2020; Huang et al., 2021; Ansmann et al., 2017). Ryder et al. (2019) estimate that by not representing these particles, dust mass over the Sahara in GCMs is underestimated by up to a factor of 5. The lack of representation of coarser dust particles in GCMs means that they may simulate a direct radiative effect (DRE) forcing that is too small in the longwave (positive DRE) and too negative in the shortwave (negative DRE) (Kok et al., 2017; Adebiyi and Kok, 2020), and therefore are too negative in total forcing (shortwave plus longwave). By representing particles up to 20 $\mu$m, Adebiyi and Kok (2020) estimate that the dust DRE at the TOA in AeroCom models (currently in the range of $-0.78$ to $-0.03$ W m$^{-2}$) would be shifted to approximately $-0.4$ to $+0.3$ W m$^{-2}$, meaning that dust could have a net warming or cooling impact on climate.

By comparing observations to model simulations, previous studies have been able to evaluate the representation of dust size distribution at various points throughout the dust life cycle. Ansmann et al. (2017) found that several dust numerical weather prediction (NWP) forecasts were accurate up to 2000 km west of the coast of Africa, but beyond this, rapid dust removal reduced the quality of the forecast in terms of the total dust mass concentration and 500-550 nm extinction coefficient. Dust-related processes in models are often tuned so that the modelled aerosol optical depth (AOD) matches observed AODs retrieved by satellite instruments. O'Sullivan et al. (2020) show that observations from a campaign obtaining in-situ and remote sensing measurements over the Eastern Atlantic agreed with an NWP forecast and a reanalysis output in terms of the AOD, but strug-

gled to show the correct vertical and horizontal distribution of coarser particles. By tuning models to AOD, a fine bias is often created in the dust size distribution to compensate for the under-represented (or absent) coarser particles.

Some studies have shown that altering certain fixed parameters in the model, such as settling velocity or particle density, can improve model agreement with observations. Drakaki et al. (2022) found that decreasing the settling velocities of dust in the model by 40-80% produced good agreement of the size distribution with in-situ aircraft observations over the Sahara and the Eastern Atlantic. By reducing the settling velocity (by 13% in line with suggestions by Huang et al. (2020)) and lowering the dust particle density from $2500\,\mathrm{kg\,m^{-3}}$ to between $125\text{-}250\,\mathrm{kg\,m^{-3}}$, Meng et al. (2022) were able to improve model agreement

with observations in terms of the super-coarse particle volume near the Sahara, though dust volume was still underestimated in dust outflow regions. These significant, order of magnitude changes to particle density and settling velocity are not representative of realistic uncertainties in these variables or processes, and instead act as a proxy to representing poorly understood processes which can potentially impact particle lifetime, such as electric charging (van der Does et al., 2018; Toth III et al., 2020; Renard et al., 2018; Méndez Harper et al., 2022), asphericity (Huang et al., 2021, 2020; Mallios et al., 2020; Saxby et al.,

2018; Colarco et al., 2014; Yang et al., 2013), turbulence (Denjean et al., 2016; Rodakoviski et al., 2023), topography (Heisel et al., 2021; Rosenberg et al., 2014) and vertical mixing (Gasteiger et al., 2017; Cornwell et al., 2021). Nowottnick et al. (2010) found that an improvement of wet scavenging processes in a model improved coarse particles lifetime.

    The Fennec (Ryder et al., 2013b, a, 2015), AERosol Properties - Dust (AER-D) (Ryder et al., 2018) and Saharan Aerosol

Long-Range Transport and Aerosol-Cloud-Interaction Experiment (SALTRACE) (Weinzierl et al., 2017) airborne campaigns measured vertically resolved size distributions at four locations between the Sahara and Caribbean and thus represent observations at different stages in the long range trans-Atlantic transport of Saharan dust. These campaigns measured the full size range of lofted mineral dust particles using open-path wing probes, unlike many previous campaigns which assumed the transport of coarser particles to be minimal, and therefore did not measure substantially into the coarse, super-coarse or giant size range,

or measurements of coarser particles were restricted by sampling constraints due to instrument inlets and pipework (Ryder et al., 2019; Rosenberg et al., 2014). This study is the first time that these three campaigns will have been analysed together, in particular taking the vertical distribution of dust size into account. In order to better understand the ability of models to simulate dust transport and deposition, these campaigns will be analysed and compared to a Met Office Unified Model (MetUM) climate simulation (HadGEM3-GA7.1) (Walters et al., 2019). HadGEM3-GA7.1 includes representation of coarse dust

particles up to 63.2 $\mu$m in diameter; a notably larger upper size limit than other models which tend to cut off the represented dust size distribution at ~20 $\mu$m (Mahowald et al., 2014; Zhao et al., 2022; Huneeus et al., 2011). The HadGEM3-GA7.1 dust simulation has not yet been extensively compared with in-situ airborne observations. The campaigns and model have not had their vertically resolved dust size distribution evolution assessed in such detail before and over such a large spatial extent, representing the vertically resolved size distribution evolution over long range transport. O'Sullivan et al. (2020) suggest that

the earlier MetUM NWP GA6.1 configuration (notably different with dust represented by two size bins) often places dust too

low in the atmosphere, over the Eastern Atlantic, which we investigate in this study.

This study aims to gain a more in-depth insight into the systematic biases between modelled and observed size distributions and how those biases evolve during transport. Such assessments of model performance are crucial in guiding improvements to model representation of mineral dust transport and deposition.

In Sect. 2, we introduce the aircraft campaigns, the model setup used in this study and our methodology for the analysis. In Sect. 3, we investigate the relationship between the coarser dust size distribution and the AOD in the aircraft observations. In Sect. 4, we present and discuss our results analysing the vertical dust structure, size distribution and concentration evolution across the Atlantic in the model and observations. In Sect. 5 we summarise and present conclusions.

## 2 Methods

### 2.1 Aircraft Observations

The vertically resolved in-situ aircraft observations used in this study were taken during scientific flights at the Sahara, Canary Islands, Cape Verde and Caribbean during the Fennec, AER-D and SALTRACE campaigns. Figure 1 shows the location of the observations (flight tracks) used in this study. All aircraft observations are presented at ambient conditions. The Fennec and AER-D campaigns made use of the BAe-146 Facility for Airborne Atmospheric Measurements (FAAM) aircraft and instruments (Ryder et al., 2013b, a, 2018), while the SALTRACE campaign used Falcon Deutsches Zentrum für Luft- und Raumfahrt (DLR) aircraft and instruments (Weinzierl et al., 2017). The following two sections describe these two different aircraft and instrumentation setups. Henceforth all aerosol sizes will be given in diameters.

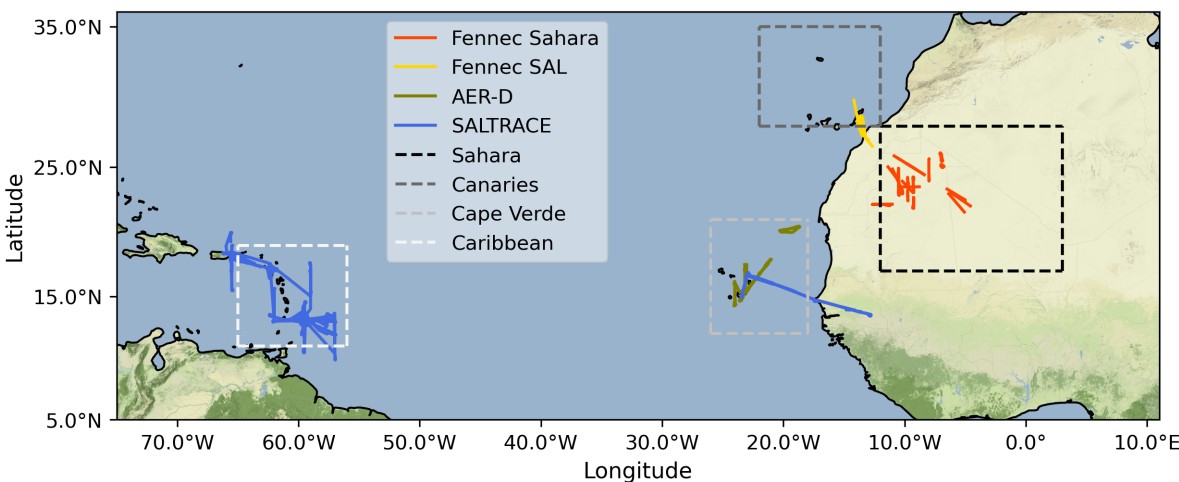

**Figure 1.** Location of the vertical profiles measured during the Fennec and AER-D campaigns, as well as the flight paths followed during the SALTRACE campaign (solid lines), and box regions used for analysis of the model data (dashed lines).

### 2.1.1 FAAM BAe-146 aircraft setup

The Fennec campaign took place in June 2011, flying over a remote region of the Sahara Desert (Mauritania and Mali), as well as over the Canary Islands (Figure 1; Fennec Sahara and Fennec SAL, respectively). This campaign therefore provides data at two separate locations; firstly over the desert close to dust sources (Fennec Sahara) (Ryder et al., 2013b) and secondly as the Saharan Air Layer (SAL) forms over the marine boundary layer (MBL) between the west coast of Africa and the Canary Islands (Fennec SAL) (Ryder et al., 2013a). In total, 41 vertical profiles were conducted during the Fennec campaign: 20 at the Canaries and 21 over the Sahara (Table 1). These profiles are conducted as the aircraft ascends/descends between the minimum safe altitude (around 160 m above ground level depending on visibility) and up to 8 km. The profiles at the Canaries were measured as the aircraft travelled to and from Fuerteventura airport (28.4°N 13.8°W) and the Sahara, so two profiles were usually measured per flight. The lowest portion of the profile was over the ocean, while the highest altitude of the profile lies just over the continent.

The AER-D campaign took place in August 2015, conducting 26 vertical profiles in the Cape Verde region. The flights from which these profiles are taken are described in Table 2.

Table 3 shows the instruments operated in each campaign and the size range applied from each instrument, adjusted from geometric to optical diameter (see Ryder et al. (2013a, 2018) for details). Both the Fennec and AER-D campaigns measured particles up to 300 $\mu$m diameter. In order to tailor our analysis to the model, only observations corresponding to the model size

| Flight number | Date | Time of flights (UTC) | Number of profiles |
|---|---|---|---|
| b600 | 17 June 2011 | 10:00-12:30 | 1C 1Ma 1Mu |
| b601 | 17 June 2011 | 15:00-19:30 | 2C 1Ma 1Mu |
| b602 | 18 June 2011 | 08:30-12:30 | 2C 1Ma 1Mu |
| b604 | 20 June 2011 | 13:00-17:30 | 2C 2Mu |
| b605 | 21 June 2011 | 10:00-12:00 | 1C 2Mu |
| b606 | 21 June 2011 | 14:00-19:00 | 2C 1Mu |
| b609 | 24 June 2011 | 11:30-16:30 | 2C 1Mu |
| b610 | 25 June 2011 | 07:30-12:00 | 2C 2Mu |
| b611 | 25 June 2011 | 14:30-19:00 | 2C 2Mu |
| b612 | 26 June 2011 | 07:30-12:00 | 2C 2Mu |
| b613 | 26 June 2011 | 14:00-18:00 | 2C 3Mu |

**Table 1.** Details of the Fennec flights used in this study including date and time of flights. Time is given to nearest 30 minutes. The number of profiles are described by the number taken from the Canaries (C) and the number at North Mali (Ma) and North Mauritania (Mu). Data taken from: Ryder et al. (2013b, a).

| Flight number | Date | Time of in-situ sampling (UTC) | Number of profiles |
|---|---|---|---|
| b920 | 7 Aug 2015 | 15:00-17:00 | 7 |
| b924 | 12 Aug 2015 | 15:30-16:30 | 1 |
| b928 | 16 Aug 2015 | 15:30-16:30 | 6 |
| b932 | 20 Aug 2015 | 11:00-12:00 | 6 |
| b934 | 25 Aug 2015 | 15:00-17:45 | 6 |

**Table 2.** Details of the AER-D flights and the times of in-situ sampling used in this study.

bins (up to 63.2 $\mu$m diameter) are used in this study.

During Fennec, wing-mounted (i.e. with no fuselage inlet) optical particle counter (OPC) probes were operated to measure the accumulation mode and coarse to super-coarse mode size distributions (passive cavity aerosol spectrometer probe (PCASP) and Cloud Droplet Probe (CDP), respectively), while measurements from a wing-mounted optical array probe (OAP), the cloud imaging probe (CIP15), are used for the super-coarse and giant modes. The OPCs use light scattering measurement techniques, and therefore the size bins applied are adjusted for a dust refractive index of 1.53-0.001i, based on scattering and absorption

measurements (Ryder et al., 2013a). Errors due to uncertainties and oscillations in the Mie scattering curve for the OPCs, in addition to systematic error for the PCASP and random (counting) errors for all probes were propagated through to size distribution uncertainties. Full details of Fennec instrument processing are given in Ryder et al. (2013b, a).

| Instrument | Abbreviation | Size range ($\mu$m) | Fennec | AER-D | SALTRACE |
|---|---|---|---|---|---|
| Passive cavity aerosol spectrometer probe 100-X | PCASP | 0.13-3.83 | Y | Y | N |
| Cloud Droplet Probe | CDP | 2.86-20 | Y | Y | N |
| Cloud Imaging Probe | CIP15 | 15-63.2 | Y | N | N |
| Two-dimensional stereo probe | 2DS | 20-63.2 | N | Y | N |
| TSI Integrating Nephelometer 3563* | Nephelometer* | < 3 | Y | Y | N |
| Radiance Research Particle Soot Absorption Photometer* | PSAP* | < 3 | Y | Y | N |
| Ultra High Sensitivity Aerosol Spectrometer | UHSAS-A | 0.08-3 | N | N | Y |
| Grimm Sky OPC | SkyOPC | 0.3-3 | N | N | Y |
| Cloud and Aerosol Spectrometer with Depolarization Detection | CAS-DPOL | 0.5-50 | N | N | Y |

**Table 3.** Size distribution instruments and scattering and absorption instruments used during the Fennec and/or AER-D campaign, where Y/N indicates instrument operation/not-operational. Sizes are given as geometric diameter. Size ranges correspond to data selected for model intercomparisons (as opposed to the full range measured by the instruments). * Indicates an instrument is located in-cabin, behind an inlet. Additional details are provided in the supplementary material in Table S1. Data taken from: Ryder et al. (2013b, 2018, 2015), Walser (2017) and Weinzierl et al. (2017).

During AER-D, the same wing-mounted OPCs were operated (PCASP and CDP), while measurements from the OAP two-dimensional stereo probe (2DS) are used for the super-coarse to giant mode. As with Fennec, the size bins applied to the OPC data are adjusted for a dust refractive index of 1.53-0.001i based on scattering and absorption measurements (Ryder et al., 2018). Sizing for the 2DS is performed using the mean of the x and y dimensions of each particle image, in order to be consistent with Fennec data processing, and is also curtailed at 300 $\mu$m for this reason, though few particles approaching this size were detected during AER-D. We propagate errors in size and number distribution due to uncertainties and oscillations in the Mie scattering curve for the OPCs, in addition to random errors (from counting and discretization error) and systematic errors (from sample area) for all instruments. Full details of AER-D instrument processing are given in Ryder et al. (2018).

During both Fennec and AER-D, the aircraft measured scattering coefficient with a TSI integrating nephelometer 3563 and absorption coefficient with a Radiance Research particle soot absorption photometer (PSAP) (Ryder et al., 2015). These instruments are located in-cabin, behind Rosemount inlets with an estimated 50% efficiency for diameters below 3 $\mu$m resulting from inlet losses and pipework transmission losses (Ryder et al., 2013b, 2018). The sum of scattering and absorption provides extinction; this has been integrated vertically to provide AOD at 550 nm, representing AOD for d < 3 $\mu$m. AOD at the time of observation could therefore be marginally larger than the AODs presented here.

Due to dust-induced visibility reductions impacting the minimum safe altitude for flying, the minimum height of observational data at the Sahara varies by flight, from around 100-500 m above ground. Therefore, we impose a minimum altitude threshold of 500 m here for the Fennec Sahara profile analysis to avoid sampling bias across different flights, weather and dust conditions. Data collected in the MBL may contain contaminated dust and non-dust aerosols, such as sea salt and anthropogenic pollution. Compositional analysis carried out by Ryder et al. (2018) on the aerosols measured during the AER-D

campaign, showed that particles d > 0.5 $\mu$m were dominated by alumino-silicates and quartz, while between 0.1-0.5 $\mu$m, the dominant particles were sulphates and salts. As we are most interested in the coarser dust particles in this study, these fine-sized contaminants should not impact our analysis. Therefore, profiles over the Canary Islands and during AER-D are analysed to their minimum sampling altitude (either ~16 m or to landing at Fuerteventura airport). Finally, filtering of the data removed noise based on a signal to noise ratio as a function of diameter.


### 2.1.2   Falcon DLR aircraft setup

The SALTRACE campaign took place in June and July 2013, conducting flights in the East Atlantic in the Cape Verde region (SALTRACE-E) and in the West Atlantic around the Caribbean (SALTRACE-W) (Figure 1 and Table 4).

During SALTRACE, the Falcon DLR took measurements using a combination of OPCs: Grimm Sky OPC (SkyOPC), Ultra High Sensitivity Aerosol Spectrometer (UHSAS-A) and the Cloud and Aerosol Spectrometer with Depolarization Detection (CAS-DPOL). Some details of these instruments are shown in Table 3. Full details can be found in Walser (2017) and Weinzierl et al. (2017) supplementary material.

We use data from both vertical profiles and horizontal segments in our analysis of the SALTRACE data. SALTRACE data from horizontal flight legs are broken down into 330 flight segments, each lasting for 150 seconds. These have been inverted and represented using lognormal modes in order to consistently propagate measurement uncertainties (e.g. optical particle counter response and properties, correction for refractive index) (Walser, 2017). These horizontal segments provide size distributions at a high resolution in diameter space. Additionally, in order to provide a vertically continuous description of dust mass and size

variation with altitude, we use SALTRACE profile observations. The profile data has not undergone such extensive processing as the horizontal segments, and instead adjustments to the instrument bin sizes were applied to account for refractive index. Comparisons between the detailed size distributions from horizontal segments and those from profiles shows good agreement (not shown). This allowed 44 size-resolved vertical profiles from SALTRACE to be analysed.

In order to calculate AOD, retrieved mass concentration profiles calculated from size distributions were combined with a mass extinction efficiency determined from an optical model (Gasteiger and Wiegner, 2018). This produced profiles of extinction coefficient which were vertically integrated to provide AOD at 500 nm. See Wieland et al. (2024) for details. The

| Flight number | Date | Location | Time of measurements (UTC) | Number of segments / full profiles |
|---|---|---|---|---|
| 130611b | 11 Jun 2013 | La Palma (ES) to Sal (CV) | 12:51-16:25 | 17 / 2 |
| 130612a | 12 Jun 2013 | Sal to Dakar (SN) | 08:52-12:08 | 19 / 2 |
| 130612b | 12 Jun 2013 | Dakar to Sal | 13:12-16:10 | 13 / 2 |
| 130614a | 14 Jun 2013 | Sal to Dakar | 09:06-12:37 | 29 / 2 |
| 130614b | 14 Jun 2013 | Dakar to Sal | 13:47-15:54 | 27 / 2 |
| 130617a | 17 Jun 2013 | Sal to Praia (CV) | 11:06-12:27 | 17 / 2 |
| 130620a | 20 Jun 2013 | Barbados | 12:01-15:55 | 32 / 2 |
| 130621a | 21 Jun 2013 | Barbados | 18:32-22:01 | 36 / 2 |
| 130622a | 22 Jun 2013 | Barbados | 18:05-21:55 | 33 / 2 |
| 130626a | 26 Jun 2013 | Barbados | 23:25-03:15 | 10 / 2 |
| 130630a | 30 Jun 2013 | Barbados to Antigua | 13:03-16:28 | 10 / 2 |
| 130701a | 1 Jul 2013 | San Juan (PR) to Antigua | 14:22-18:12 | 16 / 4 |
| 130701b | 1 Jul 2013 | Antigua to Barbados | 19:48-23:30 | 12 / 4 |
| 130705a | 5 Jul 2013 | Barbados | 12:10-16:01 | 0 / 2 |
| 130708a | 8 Jul 2013 | Barbados | 18:55-22:46 | 0 / 4 |
| 130710a | 10 Jul 2013 | Barbados | 15:07-19:18 | 25 / 4 |
| 130711a | 11 Jul 2013 | Barbados | 12:37-15:03 | 10 / 2 |
| 130711b | 11 Jul 2013 | Barbados to San Juan | 18:04-21:05 | 24 / 2 |

**Table 4.** Details of the SALTRACE flights, including location, and the time (UTC) of flights. Where ES is Spain, CV is Cape Verde, SN is Senegal and PR is Puerto Rico. The number of horizontal segments and vertical profiles measured during each flight are shown; each horizontal segment is measured over 150 seconds. Data taken from: Weinzierl et al. (2017) supplementary material.

SALTRACE AODs therefore represent the full size range in contrast to those which use the FAAM data.

Atmospheric concentrations of coarse and super-coarse particles during the airborne measurements of the presented mean vertical mass concentration profiles were often near to or below the detection limit of the CAS-DPOL. Hence, the mean mass concentrations should be considered as a lower threshold.

### 2.1.3   Processing of aircraft data

For all campaigns, profile data was aggregated across instrument size bins to match the broader six size bins of the model (Table 5), assuming homogeneous distributions across instrument size bins. For example, for Fennec Sahara, model size bin 1 is compared against corresponding data at sizes measured by the PCASP ($0.0632 \leq$ d $< 0.2$ $\mu$m) while model size bin 6 ($20 \leq$ d $< 63.2$ $\mu$m) is compared against data from the CIP15. Where model and instrument size bins did not match up perfectly, number concentration was proportioned across instrumental size bins. For example, for SALTRACE, model size bin

4 ($2 \leq d < 6.32$ $\mu$m) is compared against concentrations measured by the CAS-DPOL over instrumental size bins 11 to 15 plus half of the number concentration from bin 10 (see supplement for full details; Table S1). This provides measured number concentrations corresponding to each model size bin as a function of time for the aircraft data. Assuming the density of dust to be 2.65 g cm$^{-3}$ (Hess et al., 1998) and that the particles are spherical, we calculate mass concentrations for each of these size bins using standard volumetric and mass equations, based on the instrumental mid-bin diameter. These size and time resolved

mass concentrations can then be manipulated as follows to provide mass concentration profiles and size distributions.

     Profiles are either measured as one single 'deep' profile, or several smaller profile segments combined together. Quasi-vertical profile data are averaged over 50 m intervals for high resolution analysis and model evaluation, for both FAAM and DLR measurements. For size distribution analysis, FAAM (i.e. Fennec, AER-D) aircraft profiles were averaged over 500 m

altitude intervals. DLR (i.e. SALTRACE) size distributions were taken from horizontal flight segments, and measurements performed within 500 m altitude bands were averaged. The data is regionally averaged for each campaign. In some portions of our analysis, we do not analyse data below 1 km or above 6 km in order to avoid the observed data becoming skewed by non-dust particles in the MBL or at the top of/above the SAL.

A caveat of our analysis is that this removes any measured particles outside the model limits ($0.063 < d < 63.2$ $\mu$m). Particles larger than 63.2 $\mu$m accounted for 10-40% of the total dust mass measured at the Sahara below 5 km, but at the Canaries and Cape Verde, these particles accounted for less than 10% of the dust mass and only occurred below 2 km (not shown). Hence, these giant particles were not included in this study as we focus our comparison on the size range transported in the model's atmosphere. Particularly over the Sahara, giant dust particles are likely to be omitted by model simulations and the extent of

this should be addressed in the future but is not in the scope of this study.

## 2.2    Model setup

The GA7.1 atmosphere-only version of the Hadley Centre Global Environment Model 3 (HadGEM3-GA7.1) (Walters et al., 2019) configuration of the MetUM is used to model, among other variables, global mineral dust concentrations and aerosol

optical depths. This setup is identical to those used in the HadGEM3 CMIP6 (Coupled Model Intercomparison Project phase 6) AMIP simulations which is configured to use observed sea surface temperatures (SSTs) and CMIP6 historical inventories (Eyring et al., 2016). The model has a horizontal grid resolution of 1.875° x 1.25° (N96), and 85 height levels, 50 of which are concentrated below 18 km. The finest vertical resolution is the lowest layer, with a depth (dZ) of 36 m. dZ increases with altitude so that at ~500 m altitude, dZ is 120 m, at ~2 km altitude, dZ is 226 m and at ~5 km altitude, dZ is 373 m. The relatively

high vertical resolution suggests that sensitivity to vertical numerical diffusion is unlikely to be important, though this may have a small effect (Zhuang et al., 2018). Mineral dust is represented by the Coupled Large-scale Aerosol Simulator for Studies in Climate (CLASSIC) scheme, described in Woodward et al. (2022), Woodward (2001) and Johnson et al. (2019). The CLASSIC dust emission scheme calculates horizontal flux in nine size bins between 0.0632 and 2000 $\mu$m diameter, and uses this to derive

vertical flux in six size bins up to 63.2 $\mu m$. Dust emissions are calculated interactively each timestep from modelled fields of friction velocity, soil moisture and the soil particle size distribution together with the model's land surface and vegetation data. A fraction of the coarsest particles are re-deposited to the surface within the same timestep as they are emitted, and these never enter the model atmosphere. The remaining particles are lofted into the atmosphere and are transported as independent tracers corresponding to the six size bins shown in Table 5. The dust scheme is called at every model time step, using the driving fields calculated directly from HadGEM3-GA7.1 and Joint UK Land Environment Simulator (JULES) (Woodward et al., 2022). The dust is mixed externally with other aerosols, which are simulated by the United Kingdom Chemistry and Aerosols (UKCA) Global Model of Aerosol Processes (GLOMAP-mode) scheme (Bellouin et al., 2013). The dust cannot act as cloud condensation nuclei or ice nucleating particles or chemically interact with the model. The dust interacts with the rest of the model through radiative interactions with the atmosphere and ocean biogeochemistry via the Model of Ecosystem Dynamics, nutrient Utilisation, Sequestration and Acidification (MEDUSA). The dust particles are also assumed to be spherical.

**Table 5.** Size range and representative diameter ($D_{rep}$) of the modelled transported mineral dust size bins in the CLASSIC aerosol scheme described in Woodward (2001) and Johnson et al. (2019). $D_{rep}$ is used in calculating the emitted size distribution and the particle settling velocity. Within each size bin, $dV/dlog(r)$ is assumed constant, where $V$ is particle volume and $r$ is particle radius.

| Bin number | Bin diameter range ($\mu$m) | Representative diameter ($\mu$m) |
|:---:|:---:|:---:|
| 1 | $0.0632 \leq d < 0.2$ | 0.112 |
| 2 | $0.2 \leq d < 0.632$ | 0.356 |
| 3 | $0.632 \leq d < 2$ | 1.12 |
| 4 | $2 \leq d < 6.32$ | 3.56 |
| 5 | $6.32 \leq d < 20$ | 11.2 |
| 6 | $20 \leq d < 63.2$ | 35.6 |

The dust emission scheme is described in detail in Woodward et al. (2022). The method of calculating horizontal ($G$) and vertical flux ($F$) is derived from the work of Marticorena and Bergametti (1995), using dry threshold friction velocities ($U_t^*$) from Bagnold (1941) with correction for soil moisture, based on the method of Fecan et al. (1998), and clay fraction ($F_c$). Measurements from Gillette (1979) are used to relate $G$ and $F$ by assuming a clay content of less than 20%. $G$ is calculated in each of the 9 size bins, $i$, representing the horizontal flux

$$G_i = \rho B U^{*3}(1 + \frac{U_{ti}^*}{U^*})(1 - (\frac{U_{ti}^*}{U^*})^2)\frac{M_i C D}{g} \tag{1}$$

where $\rho$ is the air density, $B$ is the bare soil fraction in the grid box, $U^*$ is the surface layer friction velocity, $M_i$ is the ratio of dust mass in the size division $i$ to the total mass, $C$ is a constant of proportionality, $D$ is a dimensionless tunable parameter and $g$ is the acceleration due to gravity. The ratio of $U^*$ to $U_t^*$ and $M$ combine to calculate the emitted size distribution, with $U_t^*$ being dependent on particle size using $D_{rep}$ values from Table 5. $M$ is calculated from the soil clay, silt and sand fractions.

The total vertical flux ($F$) is represented with six size bins. The mass in each is related to the total horizontal flux across all nine size bins (Woodward et al., 2022) according to:

$$F_i = 10^{(13.4F_c - 6.0)} G_i \frac{\Sigma_{i=1,9}(G_i)}{\Sigma_{i=1,6}(G_i)} \tag{2}$$

The particles are then transported as six independent tracers and are subject to deposition by below-cloud scavenging, gravitational settling and turbulent mixing in the boundary layer (BL). The impact of gravitational settling on the distribution of dust mass is calculated by computing the flux of dust out of a given layer and down to up to two model levels below (determined partly by the vertical spacing of the model levels), in proportion to the stokes velocity and the length of the timestep. The sensitivity of model results to the precise numerics have not been tested. Dry deposition in the BL is calculated using a resistance analogue method where the particle deposition velocity is treated as an inverse resistance based on gravitational settling and turbulent mixing (Seinfeld, 1986).

The model dust emissions are tuned to improve agreement between the simulation and observations of AOD, near-surface concentrations and deposition rates. To do this, three dimensionless parameters are altered: a global emissions multiplier, a friction velocity multiplier, and a soil moisture multiplier. The purpose of tuning is to correct for the effects of processes not included in the model, such as gustiness of wind at the source and the relationship of soil moisture in the model's top level and at the soil surface (Woodward et al., 2022). The dust was not specifically tuned for this study and an improved dust simulation would almost certainly be achievable if tuning were undertaken. However, we chose to use this configuration of settings as it is the same as those used in the HadGEM3 CMIP6 AMIP simulations (Eyring et al., 2016) and has been widely used.

The model is free-running, but uses observed SSTs to simulate five June months, 2010-2014, which outputs vertically resolved daily mean dust mass mixing ratios for each size bin. The averaged five Junes provide a 'June climatology' which is used to compare with our campaign averages. As the model is free-running, it does not represent specific meteorology and dust events, and therefore we cannot compare the specific dates on which the measurements were taken. We found minimal variability in Moderate Resolution Imaging Spectroradiometer (MODIS) Terra AOD in the two adjacent five year periods (2005-2009 and 2015-2019), suggesting that this five year period captures relatively average conditions and is of sufficient length for this study. The data is averaged over boxes representative of the campaign locations (Figure 1). Careful consideration was taken to make sure that the boxes were suitably located so as to represent the locations measured during the observations. The Sahara, Canaries and Cape Verde boxes do not overlap with the African coast as this was found to alter the distribution and magnitude of the vertical dust profile.

The daily mean dust mixing ratio, temperature and pressure on model levels are used to calculate the air density, and the mass, number, volume and surface area concentration per size bin. The calculations of the size distributions and normalisations

were carried out in the same way as with the aircraft data. The model data is not averaged in the vertical.

## 3  Confirming representivity of the aircraft observations

As shown in Tables 1, 2 and 4, the aircraft campaigns cover limited periods of time, often only taking measurements for two to three weeks. The data collected during these campaigns can be biased towards certain types of events, for example, an effort may be made to schedule and direct flights through forecasted high concentration dust events. Assuming that there may have been a scheduling bias towards high concentration dust events during the campaigns, it is important to understand to what extent the dust size distribution, especially at the coarser size range, is dependent on the AOD, which we are using to represent the magnitude of the dust event. In this section, we show that any bias in data collection is unlikely to impact the findings from this study.

### 3.1  Spatial AOD comparisons

In order to ascertain whether the dust conditions measured during the campaigns are representative of average conditions, combined MODIS dark target and deep blue AOD retrievals for land and ocean (Levy et al., 2013; Hsu et al., 2013) from the Terra satellite are used. A monthly mean AOD at 550 nm during the campaigns (June 2011 for Fennec, June 2013 for SALTRACE and August 2015 for AER-D) at the campaign locations (i.e. regional boxes shown in Figure 1) was compared to a 5 year (2010-2014) and 20 year (2000-2019) average of AOD in June (or August for AER-D). During the Fennec campaign in June 2011, the variability of the AOD at the Sahara was comparable to the longer term June averages, whereas at the Canaries, the AOD during the Fennec campaign in June 2011 was greater (AOD between 0.4-0.6) than the 5 and 20 year averages (0.2-0.4), seemingly due to a slightly more northwestwards transport of dust during June 2011. At Cape Verde during the AER-D campaign, the mean August AOD was comparable to the longer term August averages. However, in June 2013, during the SALTRACE campaign, the AOD at Cape Verde and the Caribbean was greater (0.5-1.0 and 0.3-0.6, respectively) than the longer averaging periods (0.5-0.6 and 0.3-0.4, respectively). This suggests that the campaigns observed conditions similar to (Fennec Sahara and AER-D) or dustier than average (Fennec SAL, SALTRACE-E and SALTRACE-W). Next, we analyse whether greater AOD impacts the shape of the measured coarse size distribution.

### 3.2  Relationship between AOD and size distribution

As AOD is the vertical integral of extinction caused by aerosols, which partially depends on number concentration, as well as size-varying optical properties, we expect a greater concentration of dust to coincide with a higher AOD value. We aim to test this hypothesis with our observational data and additionally, we want to understand the dependence of coarse particle size distribution on AOD; do high AOD events contain a different proportion of coarser size bin 5 and 6 (6.32-63.2 $\mu$m) particles

than low AOD events?

Here, we show the impact of AOD on size distribution by splitting campaign flights into low, medium and high magnitude AOD events based on in-situ AOD measurements taken during the Fennec (Ryder et al., 2015) AER-D and SALTRACE campaigns. The minimum, maximum and mean AOD from the AER-D campaign profiles was 0.06, 0.92 and 0.42, respectively. The AOD thresholds used to split up each campaign is given in the Table 6 caption; these thresholds were chosen as they

approximately split the number of profiles from each campaign into thirds and are different for each campaign. We use the Student's t-test to test the statistical significance of our proposed hypotheses. The smaller the returned p-value, the greater the statistical significance of the observed difference. We propose the null hypothesis states that there is no difference between the total dust mass concentration profile in low (L), medium (M) and high (H) AOD events. We found a statistically significant difference (to 95% confidence interval) between the total dust mass concentration and the AOD measured at the Sahara, Ca-

naries and Cape Verde during L, M and H events (Table 6 indicated by small p-values) in most cases; hence, we reject our null hypothesis. Thus, low AOD events measured during the two campaigns, for example, had a significantly different concentration profile to medium or high AOD events.

**Table 6.** P-values resulting from a Student's t-test to test the null hypothesis: there is no difference between the total mass concentration profile in low (L), medium (M) and high (H) AOD events. Bold values are significant to a 95% confidence interval. This is tested for the Fennec (Sahara and Canaries) and the AER-D (Cape Verde) campaign data. Each set of aircraft profiles from each location was split into thirds based on AOD at 550 nm measurements from the aircraft. The thresholds separating the low to medium, and medium to high AOD categories at the Sahara, Canaries, and Cape Verde are: 0.75 and 1.5, 0.5 and 0.75, and 0.4 and 0.6, respectively.

| Total concentration $\mu$g m$^{-3}$ | | |
|---|---|---|
| Sahara | M | H |
| L | **0.014** | **4.473**$e^{-18}$ |
| M | - | **1.100**$e^{-8}$ |
| Canaries | M | H |
| L | **2.460**$e^{-7}$ | **6.837**$e^{-10}$ |
| M | - | 0.463 |
| Cape Verde | M | H |
| L | **3.167**$e^{-8}$ | **1.028**$e^{-14}$ |
| M | - | **1.110**$e^{-4}$ |

Next, we look at the relationship between the AOD and the relative mass contribution of coarse particles to the total mass

concentration at each location. Is it difficult to determine the relationship between AOD and size distribution in observations because these measurements often characterise a different subset of the full dust size range, but even qualitative insights are

worthwhile. Figure 2 shows AOD as a function of the size bin 6 mass contribution for the Fennec, AER-D and SALTRACE campaigns (see Supplementary Material for equivalent size bin 5 figure). AOD is calculated differently between the Fennec and AER-D, and SALTRACE campaigns due to different instrumentation, but no campaign individually shows a strong correlation

 between coarse mass contribution and AOD. Combining campaign data, there is a suggestion of a correlation between the AOD and coarse mass contribution, whereby coarse contribution may increase with AOD, which is to be expected in some cases as a result of different transport distances for each campaign region. So, a model bias in AOD is unlikely to be a dominant cause for simulating too few or too many coarse particles. The next Section investigates the difference in size distribution further to identify additional causes.

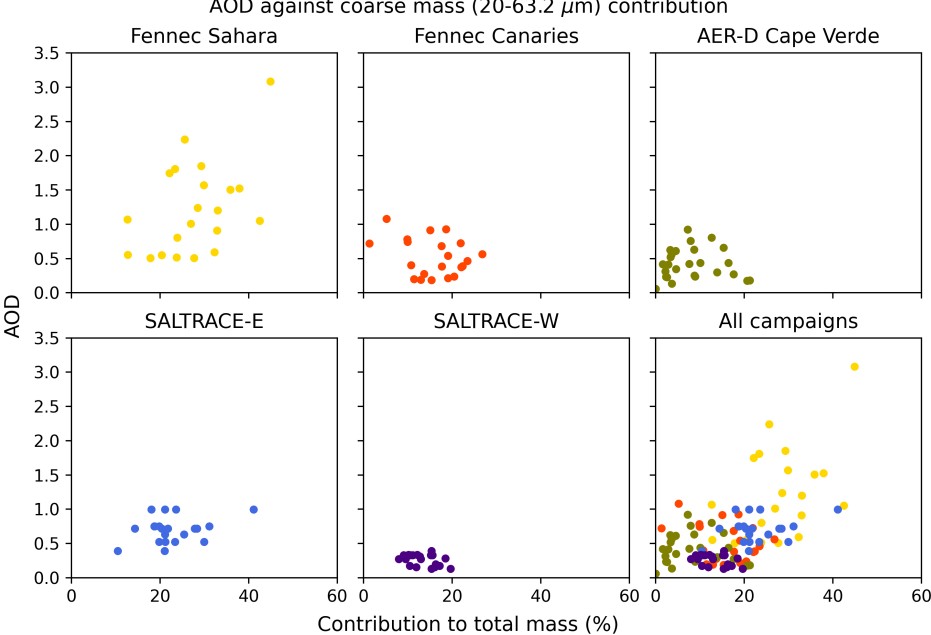

**Figure 2.** AOD against coarse mass (20-63.2 μm; size bin 6) contribution to total mass in each campaign. For Fennec and AER-D, AODs represent particles with diameters below 3 $\mu$m and mass contribution was averaged over profiles between 1-5 km at the Sahara, 0-5.5 km at the Canaries and 0-5 km at Cape Verde. For SALTRACE, AOD represents the full size range and mass contribution is taken from horizontal segments.

## 4   Results

In this section, the observations at the four observed locations (Sahara, Canaries, Cape Verde and Caribbean) will be compared to the model simulation. Initially, this comparison will investigate the specifics of the vertical structure of the dust layer before focusing on the evolution of the observed and modelled size distributions over long range transport.

## 4.1 Vertical structure

In terms of the absolute values, we have analysed the mean total mass concentration profile from each location between 0.063-63.2 $\mu$m diameter to match the modelled size range and between 1-6 km altitude to avoid contamination from the MBL or above the SAL. The mean mass concentration from observations between 1-6 km from each set of profiles has been calculated: 341 $\mu$g m$^{-3}$ at Sahara, 162 $\mu$g m$^{-3}$ at Canaries, 161 $\mu$g m$^{-3}$ and 1680 $\mu$g m$^{-3}$ at Cape Verde (AER-D and SALTRACE-E, respectively) and 340 $\mu$g m$^{-3}$ at Caribbean. Despite the expectation that the highest mean concentration would be measured at the Sahara, the SALTRACE-E mean is almost 5 times larger, while the SALTRACE-W mean is nearly as large as that measured at the Sahara. This suggests that the events measured during the SALTRACE campaign were significantly larger than those measured during Fennec and AER-D. Despite these campaigns covering a range of magnitudes, the model tends to underestimate the mean total dust mass by a factor of between 4 and 44 (not shown), with the largest underestimations occurring with the comparison to the SALTRACE-E data. It is likely that this underestimation is partly due to a bias in the model size distribution towards smaller particles which constitute less mass. This underestimation is also likely a consequence of the tuning which has been applied to the model emissions as well as the different temporal scales which we are comparing. Due to the large magnitude of difference between the model and campaigns, the vertical mass profiles have been normalised. In order to compare the vertical distribution of dust, the profiles have been normalised by the mean dust mass concentration between 1-6 km altitude.

Figure 3 shows the normalised observed and modelled vertical profiles of total dust mass concentration at each location from each campaign. Firstly, in terms of the observations, at the Sahara (Figure 3a), dust mass is highest near to the surface, likely due to the high quantity of coarse and super-coarse particles which are lofted and settle relatively close to the source. The mass concentration gradually decreases to near zero at 5.5 km, marking the top of the Saharan atmospheric boundary layer (SABL)–a well-mixed, dry layer over the Sahara extending from the surface, often up to ~6 km over the Sahara(Cuesta et al., 2009). At the Canaries (Figure 3b), the observations start to show the formation of the SAL–the dry, dusty air layer formed when the SABl rises isentropically over the Atlantic ocean's MBL (Carlson, 2016), residing between ~1-6 km–with higher concentrations of dust between 2.5-3.5 km altitude, though the profile has relatively high concentrations up to 5.5 km where it is capped at the top of the SAL. With more time and distance from the Sahara, profiles at Cape Verde (Figure 3c) represent a more mature version of the SAL; the AER-D profile has a more well-defined base and cap to the SAL with a more concentrated centre between 2-4 km. Though not as dramatic as the AER-D profile, the SALTRACE-E profile still peaks between 2-4.5 km and tails off at both the top and bottom ends of the profile. Finally, at the Caribbean (Figure 3d), the dust plume has lowered, bringing the dust mass closer to the surface and lowering the SAL cap to below 5 km.

Generally, the shape of the modelled vertical profile resembles the observed profile. However, the model has struggled to represent the rate of change of concentration with height, failing to capture the relative magnitude of the maximum and minimum values measured during Fennec and AER-D (Figure 3 a, b, and c). At the Sahara, the model represents a more well-mixed profile whereby the concentration decreases more gradually with altitude than in the observations. The model does not

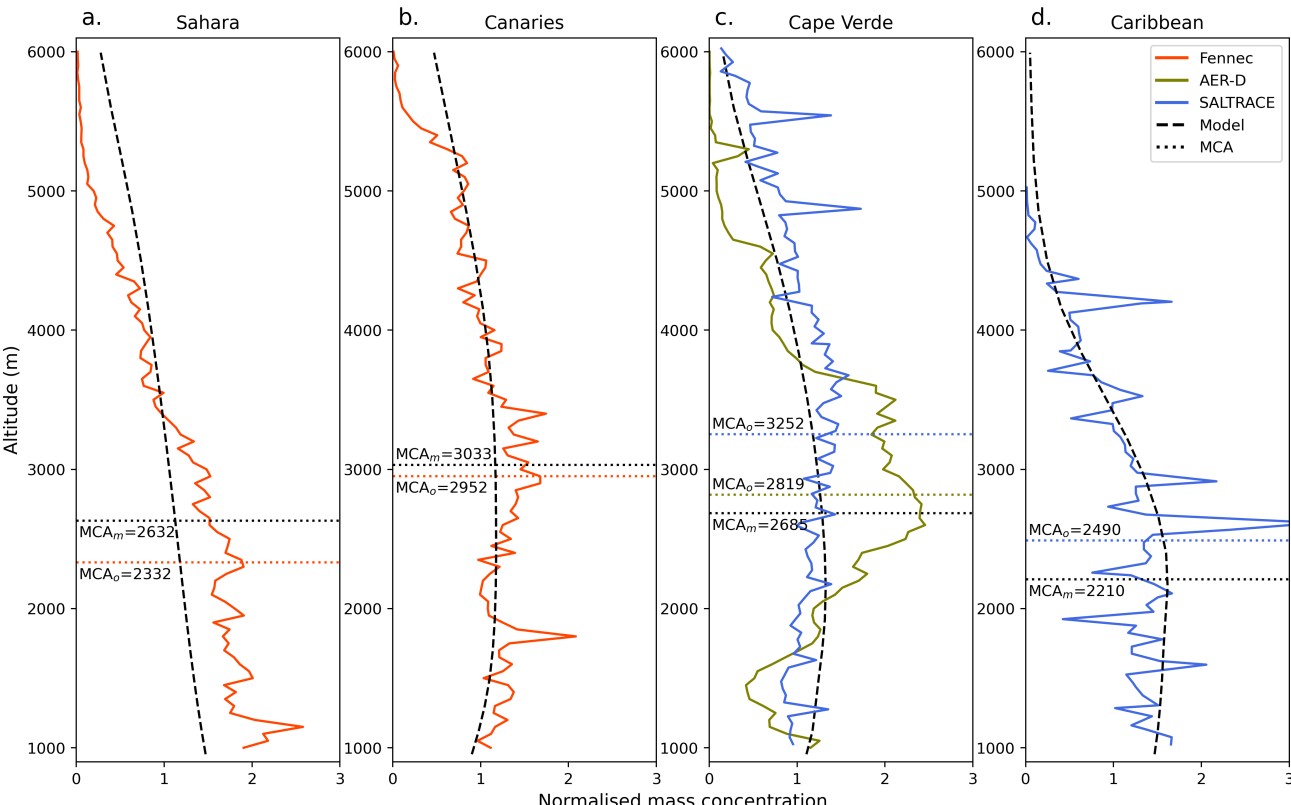

**Figure 3.** Normalised observed (coloured solid line) and modelled (black dashed line) total dust mass concentration profile and dust mass centroid altitude (MCA; dotted horizontal lines in metres) between 1-6 km. $MCA_o$ and $MCA_m$ respectively represent the observed and modelled MCA values. Plots show all four observed locations: Sahara (a), Canaries (b), Cape Verde (c; AER-D and SALTRACE-E) and Caribbean (d; SALTRACE-W) from the Fennec (orange), AER-D (green) and SALTRACE (blue) campaigns. Data has been normalised by the mean profile concentration between 1-6 km altitude.

have the same sharp cap at the top of the SABL that we see in the observations. Although the model does not represent the greater mid-SAL concentrations measured in AER-D well at Cape Verde, its vertical distribution lies fairly close to that from SALTRACE-E (Figure 3c).

The model appears to represent the top of the SAL most effectively at the Caribbean as the only location where the modelled concentration drops close to 0 at the observed SAL top. The model failing to capture this sharp decrease could be in part due to our temporal averaging of the model data, suggesting that the top of the modelled SAL could vary significantly and can occur above 6 km altitude, except for at the Caribbean. The smooth profiles could also be a consequence of limited spatial resolution

and numerical diffusion in the model.

The model represents the shape of the observed profiles very well despite the campaigns measuring fairly different total
mass concentrations. However, although the AER-D campaign measured similar mean mass concentrations at Cape Verde to
the Canaries during Fennec, the AER-D profile is the least well-fitted to the model profiles, as well as appearing fairly different
in structure to the SALTRACE-E profile. This difference could be caused by variation in the location of dust emission, which
may alter the dust size distribution and distance transported before measurement. The difference could also be a consequence
of the different time of year in which the AER-D campaign took place in; Fennec and SALTRACE both occurred in June,
whereas AER-D happened during August. The time of year impacts the location of the inter-tropical convergence zone (ITCZ)
and the strength of the Saharan Heat Low (SHL), which work together as the main cause of intense dust uplift in the early
summer (Marsham et al., 2008). The difference in meteorology could be why we see a different profile structure measured
during the AER-D campaign.

The dust mass centroid altitude (MCA) between 1-6 km – the altitude at which 50% of the mass is below and 50% is above
(Lu et al., 2023) – is shown in Figure 3. We have not included particles in the lowest 1 km of the atmosphere in our calculations
of the MCA due to potential interference from non-dust particles measured in the observations which may lower the MCA.
Hence, this value is not a total column mass, but is representative of the dust mass between 1-6 km at each location. At every
location, the modelled MCA is in a similar altitude to the observed MCA, suggesting that the model distributes the total dust
mass well in the SAL when compared to observations, in terms of the vertical distribution.

Moving away from the Sahara where the observed MCA is 2332 m, the MCA rises as the dust mass travels to the Canaries
and Cape Verde in the observations. The formation of the MBL aids in the removal of dust mass from the base of the SAL,
causing the MCA to rise; 2952 m at the Canaries and 2819 m and 3252 m at Cape Verde. Though as the plume sinks over
the West Atlantic, the MCA reduces to 2490 m at the Caribbean. This raising and lowering of the MCA across the Atlantic
is exactly what we would expect to see in our observations (e.g. Carlson (2016)). The model succeeds in representing vertical
change in the MCA across the Atlantic. We have shown that the model represents the total dust mass vertical distribution fairly
well. O'Sullivan et al. (2020) previously found that an NWP GA6.1 configuration of the MetUM placed dust 0.5-2.5 km too
low in the atmosphere when compared with observations. Our analysis of these profiles suggests that this MetUM climate
configuration may transport the dust at similar altitudes and distributions to the observations, at least in terms of the total mass
across the whole size distribution.

In order to analyse the size distribution that makes up the vertical structure at these locations, we have broken the profiles
(shown in their normalised form in Figure 3) down into the six size bins used by the CLASSIC scheme in HadGEM3-GA7.1.
We analyse the percentage contribution of mass to the total mass as a function of size. Figure 4 shows the contribution by size
bin and the mean total mass concentration from each campaign for both model and observations. Table 7 contains the mean

percentage mass contribution to total mass between 2-3.7 km altitude from the three coarsest size bins (2-6.32 $\mu$m, 6.32-20 $\mu$m and 20-63.2 $\mu$m; green, blue and purple in Figure 4) at each location from the observations and model.

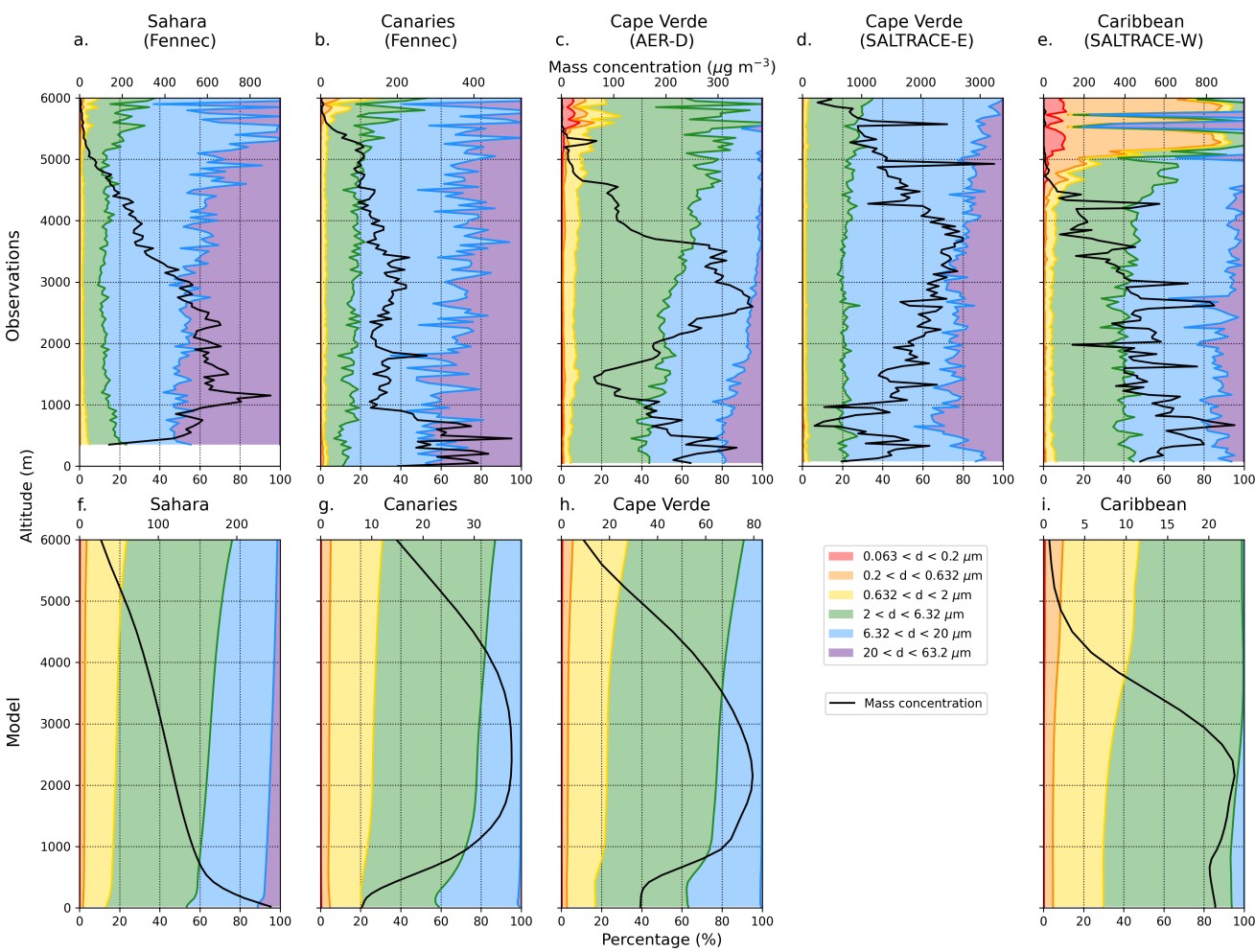

**Figure 4.** Dust mass concentration profiles, showing the total dust mass concentration in $\mu$g m$^{-3}$ (black line) and the percentage contribution of dust mass in the six model size bins (coloured areas). Plots include the mean profiles from the observations (top) and from the model (bottom) at the Sahara (Fennec; a and f), Canaries (Fennec; b and g), Cape Verde (AER-D and SALTRACE-E; c, d and h) and Caribbean (SALTRACE-W; e and i). Note that the mass concentration (black line) scales differ between panels.

At the Sahara, up to 90% of the observed dust mass up to 5 km comes from particles 6.32-63.2 $\mu$m in diameter (size bins 5 and 6; blue and purple; Figure 4a). As the dust moves westwards over the Atlantic, the contribution of these coarsest particles decreases as they are deposited from the dust plume. Between 2-3.7 km, the 6.32-63.2 $\mu$m contribution decreases from ~87% at the Sahara to ~82% at the Canaries, ~45-79% at Cape Verde and ~60% at the Caribbean (Table 7). As the coarser

contribution decreases, the contribution of 2-6.32 $\mu$m particles (size bin 4; green) increases, while the contribution of the finest

particles (0.063-2 $\mu$m; size bins 1-3; red, orange and yellow) remains low up to the top of the SAL; less than 5% at all locations except for Cape Verde during AER-D (Figure 4c). The coarse and super-coarse particles (6.32-63.2 $\mu$m; blue and purple) show a higher dependence on altitude in the AER-D data, whereby their mass contribution is highest in the lowest 1 km at up to 60% and decreases with altitude to half this contribution at 5 km. Fewer coarser particles were measured during the AER-D campaign, resulting in a higher contribution of 2-6.32 $\mu$m particles (size bin 4; green) compared to the other campaigns. The

SALTRACE-E profile (Figure 4d) shows a similar structure to the Fennec observations, suggesting that the AER-D campaign is the more anomalous of the two datasets.

**Table 7.** Mean percentage mass contribution to total mass between 2-3.7 km altitude from the three largest model size bins (2-6.32 $\mu$m, 6.32-20 $\mu$m and 20-63.2 $\mu$m; green, blue and purple in Figure 4) at the Sahara, Canaries, Cape Verde (AER-D (A) and SALTRACE-E (S)) and Caribbean in the observations (Obs) and model. Data relates to Figure 4.

| | Sahara | | Canaries | | Cape Verde | | | Caribbean | |
|---|---|---|---|---|---|---|---|---|---|
| | Obs | Model | Obs | Model | Obs (A) | Obs (S) | Model | Obs | Model |
| Bin 4; 2-6.32 $\mu$m | 10 | 46 | 14 | 52 | 48 | 19 | 55 | 35 | 64 |
| Bin 5; 6.32-20 $\mu$m | 43 | 31 | 54 | 21 | 40 | 55 | 22 | 50 | 2 |
| Bin 6; 20-63.2 $\mu$m | 44 | 4 | 28 | 0.2 | 5 | 24 | 0.1 | 10 | 9e-6 |

In general, the model overestimates the mass contribution from 0.063-6.32 $\mu$m dust particles (size bins 1-4; red, orange, yellow and green) and underestimates the 6.32-63.2 $\mu$m particle (size bins 5 and 6; blue and purple) contribution at all loca-

tions. At the Sahara, the modelled dust mass between 6.32-63.2 $\mu$m between 2-3.7 km accounts for 35% of total mass, less than half of the observed contribution of ~87% (Figure 4a and f). In the model, less than 15% of the contribution at the surface is made up of the coarsest particles (20-63.2 $\mu$m; size bin 6; purple), decreasing to ~4% between 2-3.7 km altitude, which is 11 times less than the observed contribution. The mass contribution of 20-63.2 $\mu$m particles (size bin 6; purple) in the model decreases quickly beyond the Sahara to become negligible. At the Canaries and Cape Verde, the vast majority of 20-63.2 $\mu$m

particles have been removed between 2-3.7 km, leaving a contribution of less than 0.1-0.2% from this size bin, two orders of magnitude less than the observed contribution measured during Fennec and SALTRACE (Figure 4b, c, d, g and h; Table 7). Upon reaching the Caribbean, only a very small fraction of the mass comes from the 20-63.2 $\mu$m particles (size bin 6; purple), and the 6.32-20 $\mu$m (size bin 5; blue) contribution below 1 km is less than 10% and only 2% between 2-3.7 km. The rate at which the model is losing coarse and super-coarse particles is resulting in an increasing bias of particles smaller than 6 $\mu$m and

thus an underestimation of the total dust mass remaining after long range transport.

From the Sahara, the modelled contribution of particles smaller than 2 $\mu$m (size bin 1-3; red, orange and yellow) is overesti-mated by a factor of 10, and up to 13, 3-12 and 9 at the Canaries, Cape Verde and Caribbean, respectively. This overestimation

of the fine particle mass confirms that the model shows a bias towards fine particles over coarser particles.


In the two largest size bins, the model shows a decreasing percentage mass contribution with altitude (Figure 4). At the Canaries for example, the model 6.32-20 $\mu$m mass contribution (size bin 5; blue) drops from ~30% at 1 km to ~15% at 5 km. Whereas in the observations, only the coarsest size bin shows this altitude dependence, whereby the 20-63.2 $\mu$m particle contribution (size bin 6; purple) decreases with altitude: from ~50% at 1 km to ~30% at 5 km at the Sahara and from ~25% at 1 km
to ~10% at 4 km at the Caribbean. Alternatively, the 6.32-20 $\mu$m contribution (size bin 5; blue) remains more consistent with altitude in the observations, or showing an increasing relative contribution due to the decreased contribution of the 20-63.2 $\mu$m contribution (size bin 6; purple). Thus, where the model shows an altitude dependence in the percentage mass contribution of the coarse and super-coarse dust, the observations only show this dependence only visibly affects the super-coarse mass contribution (i.e. 20-63.2 $\mu$m; purple).


The model represents the relative mass contribution of coarse and super-coarse particles as relatively height dependent, decreasing with altitude. However, the observations show little variation of coarse dust with height and a decreasing super-coarse dust contribution with height. The model fails to retain the super-coarse dust during trans-Atlantic transport and incorrectly represents the vertical distribution of coarse dust, with a bias towards lower altitudes.


## 4.2 Size distribution evolution

The height resolved modelled and observed volume size distributions have been normalised by total volume (Figure 5). This highlights the peak of the size distribution, and the difference in shape between the model and observations when the total concentrations are different. There are two things which are clear amongst all campaigns. Firstly, the shape of the distributions
from the smallest size bin to the peak in volume; the model displays a broader shape, while the observations show a more steeply curved, peaking shape. Secondly, the model underestimates volume in the largest size bin at all locations. Beginning at the Sahara, the difference is around one order of magnitude. Moving downwind, the difference between the model and observations continues to grow by orders of magnitude, such that the model volume distribution drops much more sharply to around 5 orders of magnitude less than the observations in size bin 6 (20-63.2 $\mu$m) by the Caribbean. At all locations (except
for Cape Verde during AER-D) the observed volume in the 2-6.32 $\mu$m range is very similar in magnitude to the volume in the 20-63.2 $\mu$m range (i.e. size bins 4 and 6), whereas in the model, there is a notable drop from the fourth to the sixth size bin. The increasing difference between the model and observations at the coarsest range is an indication of rapid deposition of the coarser particles in the model. Not only do we see a growing difference with distance from the Sahara, but the underestimation of coarser dust volume at the Sahara suggests there may be an issue with the model emissions and/or vertical transport whereby
not enough coarse and super-coarse particles are transported through the SABL. This underestimation is exacerbated through long range transport by the overly swift deposition of the coarser particles.

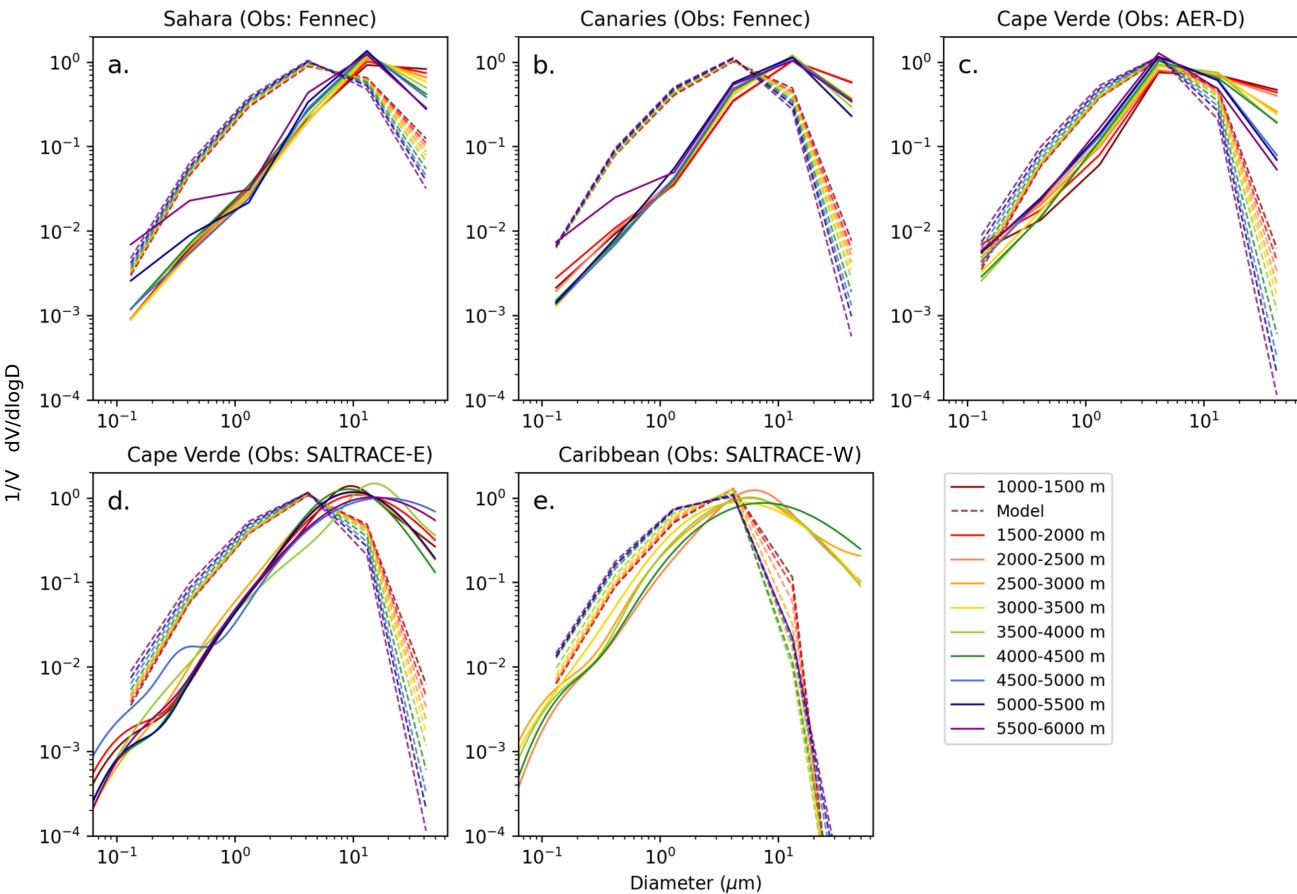

**Figure 5.** Vertically resolved modelled (dashed lines) and observed (solid lines) normalised volume distribution at the Sahara, Canaries (Fennec), Cape Verde (AER-D and SALTRACE-E) and Caribbean (SALTRACE-W) at different altitudes. The volume distributions have been normalised by the total particle volume.

The model tends to peak in volume in the 2-6.32 $\mu$m bin, whereas the observations measured during the Fennec and SALTRACE campaigns peak in the next size bin up (6.32-20 $\mu$m). Contrary to the other campaigns, the volume distribution from the AER-D campaign at Cape Verde peaks in the 2-6.32 $\mu$m bin. As mentioned previously when observing the different vertical structure, this difference could be a consequence of the different time of year in which this campaign occurred. Despite the differences between the data collected from the AER-D campaign and the Fennec and SALTRACE campaigns, the AER-D data remains consistent with the other campaigns in showing that the model underestimates coarser dust particle mass and transport.

When normalised, there is no particular pattern in relation to the altitude except for at coarsest size ranges, where dust volume tends to decrease with altitude in the model and observations. Otherwise, the shape of the distribution remains consistent with altitude. The only other exception being at high altitudes in the smallest size bin in the observations at the Sahara, Canaries and Cape Verde (Figure 5a, b and d). The volume distribution of fine 0.063-0.632 $\mu$m particles is greater above 5 km than at lower altitudes. This could be a signal of non-dust particles.

Figures 4 and 5 show that the model under represents the coarser size distribution over the Sahara, as well as further downwind during transport. In this study, we focus on the impact of transport processes on the size distribution, rather than examining emission processes. The model emitted size distribution is dominated by size bin 6, although the atmospheric dust size distribution in the lowest model level is already dominated by size bin 4 (see Supplement Figure S1). This suggests additional challenges in representing initial dust transport from emission into the very low atmosphere, which should be an area for future study. The aggregated Saharan observations presented here are from 500 m upwards, which prevents a detailed analysis of near-surface emission size distribution.

In order to illustrate how the model represents the evolution of dust size distribution during trans-Atlantic transport and the discrepancies between the model and observations over the Sahara, Figure 6 shows the vertically-resolved fractional model underestimate of the volume size distribution between the observations and the model at the Sahara, Canaries, Cape Verde and Caribbean (i.e. observations/model dV/dlogD). This Figure shows that at all locations, the model underestimates the coarse fractions by greater orders of magnitude than the fine fraction. Additionally, and most significantly, the magnitude of the coarse fraction underestimate grows with transport from the Sahara; the size bin 6 fractional underestimate increases from around a factor of 10 over the Sahara to over one million at the Caribbean. Thus, we demonstrate that although there is an underestimation of the volume distribution at the source, this is significantly exacerbated by several orders of magnitude with westwards transport.

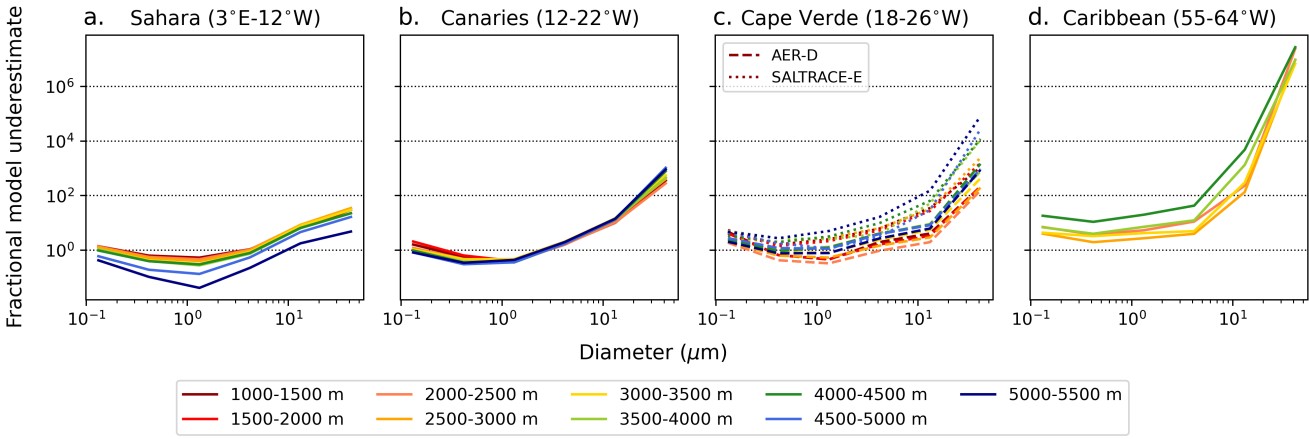

**Figure 6.** Fractional underestimate of the volume size distribution between the observations and model. The vertically-resolved difference is shown at the Sahara (a) and Canaries (b), using Fennec observational data, at Cape Verde (c) using AER-D (dashed lines) and SALTRACE-E (dotted lines) data and at the Caribbean (d) using SALTRACE-W data.

We have postulated previously that the model struggles to raise the coarse dust high enough, showing more altitudinal dependence than the observations. In Figure 6c and d at Cape Verde and the Caribbean, the model underestimate becomes worse at higher altitudes. At Cape Verde in the SALTRACE-E comparison, there is an order of magnitude difference between the underestimate of size bin 6 between 1-1.5 km and 5-5.5 km altitude.

While no observations of vertically-resolved, size-resolved dust concentration over the mid-Atlantic exist, we are able to look at how the model simulates the concentration evolution across the Atlantic. Figure 7 shows the evolution of mass concentration in each size bin from the Sahara to the Caribbean. Figure 7a shows the modelled mean June AOD as well as stippling which represents the 65th percentile of AOD between 1°S and 47°N at each longitude which has been used to identify the mean latitudinal plume extent. The 65th percentile of AOD was chosen so as to cover an area including all the observed lo-
cations. Figure 7b shows the modelled mass concentration in each of the six size bins in the defined dust plume location and between the 2-3.7 km altitude range. The 2-3.7 km altitude range has been selected to analyse the dust plume to minimise interference from the MBL and free troposphere above the SAL, across the entire Atlantic. Figure 7b shows that the 2-6.32 $\mu$m particles (green) are the dominant contributors to dust mass across the Atlantic in the model, as in Figure 4. Although the mass of 6.32-20 $\mu$m particles (blue) are double that of the 0.632-2 $\mu$m particles (red, orange and yellow) at the Sahara, these larger
particles are removed more swiftly and have less mass than the finer particles west of Cape Verde (~25°W).

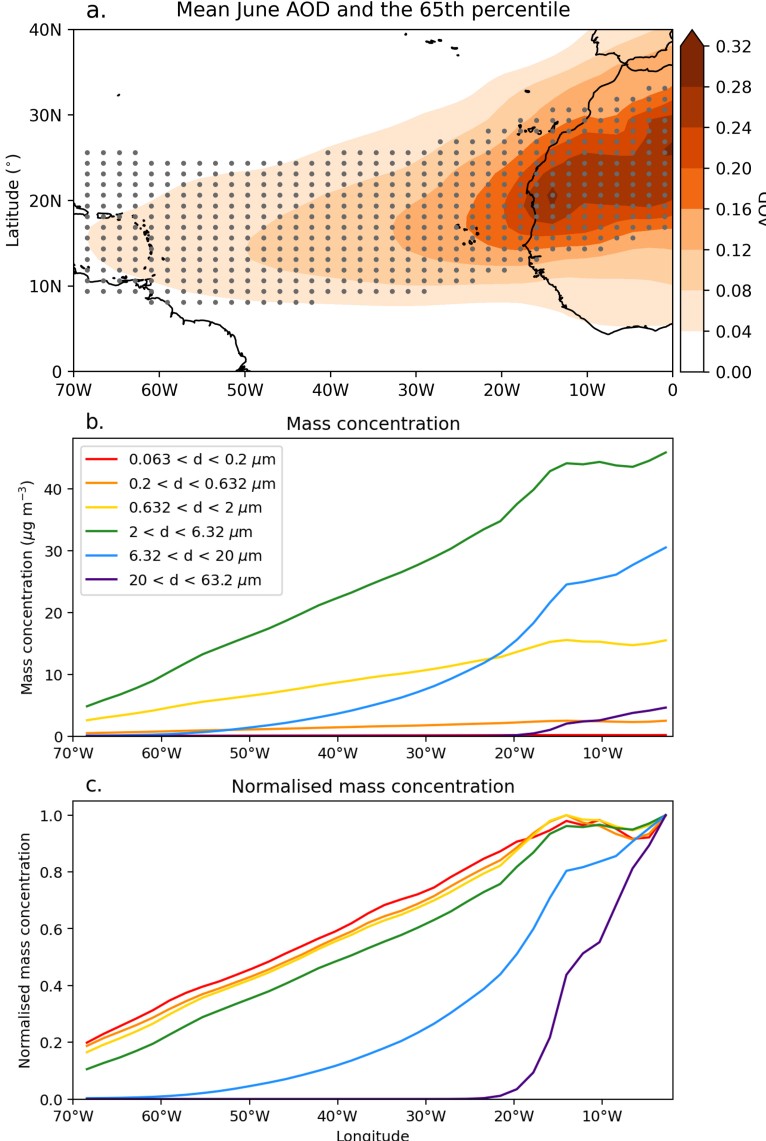

**Figure 7.** The 65th percentile (stippling) of mean June 2010-2014 AOD at 550 nm (orange shading) at each longitude has been used to locate the dust plume (a). The mean modelled dust mass concentration between 2.0-3.7 km altitude by longitude in the six CLASSIC size bins from the Sahara to the Caribbean (b). The binned concentrations have been normalised by the mass concentration in each bin at 3°W (c).

Figure 7c shows the normalised mass concentration transect in the six size bins. These have been normalised by their value at the Sahara (~3°W) to allow a direct comparison of the rate of change of mass concentration between each size bin. All size bins experience change in their mass concentration in two distinct regions, one over the African continent (15-3°W) and over
the Atlantic, where each size bin loses mass at a size-dependent rate. These two distinct areas hint at the different processes

which alter dust transport over land and ocean. The rate of loss of the four finest size bins (0.063-6.32 $\mu$m; red-green) appears fairly linear; each size bin loses 70-80% of its mass between the west African coast and the Caribbean. The rate of loss of the coarsest size bins (6.32-63.2 $\mu$m; blue and purple) is sharper. These much faster rates of loss result in a negligible mass of 20-63.2 $\mu$m particles remaining shortly west of Cape Verde and of 6.32-20 $\mu$m particles remaining near the Caribbean.

## 550  5   Conclusions

Vertically resolved, in-situ observations from three aircraft campaigns, Fennec, AER-D and SALTRACE, at the Sahara, Canary Islands, Cape Verde and Caribbean are analysed together to understand the evolution of dust particle size distribution over long range transport, with a particular focus on the coarser particles and their vertical distribution. The observations from these campaigns are used to evaluate the Met Office Unified Model (MetUM) HadGEM3-GA7.1 climate model representation of
the dust size distribution across the Atlantic. This work presents the first time that all three of these campaigns have been used together and analysed in such high vertical resolution in order to understand the size distribution evolution from the Sahara to the Caribbean, as well as being the most extensive evaluation of the MetUM HadGEM3-GA7.1 model representation of long range dust size distribution evolution.

Aircraft observations from the Fennec, AER-D and SALTRACE campaigns show that coarser particles are being transported further in the real-world than in the model, in which coarser dust particles (6.32-63.2 $\mu$m) are underestimated in both mass and volume size distribution at all stages of long range transport. At the Sahara, the model underestimates the normalized volume size distribution of the largest particles (20-63.2 $\mu$m) by more than one order of magnitude. The contribution of 20-63.2 $\mu$m particle mass to the total mass is only 4% in the model and 44% on average in the observations between 2-3.7 km over the
Sahara, resulting in a model underestimation by a factor of 11. Size bin 5 particle mass contribution is ~43% in the observations and only 31% in the model. This result in a stark overestimation of the 2-6.32 $\mu$m mass contribution by 36%. These underestimations of the coarser particles suggest a challenge in representing the immediate transport upwards through the atmosphere after emission.

Observations suggest that the contribution of coarse particle mass to total mass is not strongly correlated with AOD, at least within a given campaign. The use of campaign periods with slightly higher than average AOD could therefore contribute to the poor representation of coarse particles in the model, but is not dominant driver. The model underestimation of coarse particle concentration is so large that AOD variations within a campaign alone are not sufficient to explain the differences between model and observations. We find that the model underestimates the coarser particle volume distribution by increasing orders of
magnitude with distance from the Sahara. The normalised volume size distribution in the largest model size bin (20-63.2 $\mu$m) is underestimated by one order of magnitude over the Sahara, up to 3 orders of magnitude at the Canaries, 5 at Cape Verde and 7 at the Caribbean. This increasing disparity between the model and observations is a consequence of overly swift removal of coarse and super-coarse particles from the modelled atmosphere which is marked over the Sahara, and is exacerbated during

long-range transport. The majority of 20-63.2 $\mu$m particles have been removed from the Saharan Air Layer (SAL) shortly west of Cape Verde, contributing only 0.1% of the total mass, where the observations show this size bin contributing up to 25% of the total mass between 2-3.7 km. The model's fifth size bin (6.32-20 $\mu$m) shows a slightly slower rate of removal from the model, however still leaving a negligible concentration at the Caribbean, where the mass contributed by this size bin to the total dust mass is underestimated by a factor of 25 between 2-3.7 km. We suggest that the model is simulating far too swift deposition of particles sized larger than 6.32 $\mu$m during the full course of long range transport, leading to an increasing underestimation of dust mass with distance from the Sahara.

We have shown that the model generally agrees with the vertical distribution of total dust mass in the observations. We show that the mass centroid altitude (MCA) in the model is consistently within range of the observations. However, we find an underestimation in super-coarse volume size distribution which increases with altitude, showing that the model increasingly struggles with coarser particle representation over long range horizontal and vertical transport, and despite representing the MCA of total dust mass, transports coarser dust particles too low in the atmosphere.

Our results are subject to some limitations. Firstly, it is noted that the aircraft observations used in this study cannot be fully representative of climatic conditions due to limitations in the temporal and spatial coverage of the observations. This makes our comparisons to the model more complex as the model provides daily mean data, covering each full 24 hour period. Additionally, we find that the AER-D data has a slightly different vertical distribution of dust compared to SALTRACE-E at Cape Verde, as well as a finer size distribution in comparison to the Fennec and SALTRACE campaigns, despite having consistent instrumentation with the Fennec campaign. It is not exactly clear what causes this disparity, but it could be a consequence of the measurements being taken in August compared to the other campaigns which were conducted in June. Additionally, we note that we excluded observations for which d > 63.2 $\mu$m since this is the maximum size represented by the model, which were significant in the Sahara observations. Finally, we must consider that any biases in the model's representation of the dust vertical and horizontal distribution, as well as the size distribution, could be due to either the dust scheme or to biases in the modelled climate. There is the potential for future research into the sensitivity of coarser particle transport in the model to the numerical schemes which could provide additional valuable information to this research topic.

We have shown that the model has difficulty with representing the coarse dust size distribution from the Sahara to the West Atlantic. This is consistent with other studies which have evaluated a range of models on more restricted spatial and vertical scales (Adebiyi and Kok, 2020; Ansmann et al., 2017; O'Sullivan et al., 2020). Incorrect representation of dust size distributions in climate models will result in erroneous dust radiative effects, impacts on clouds, and deposition of nutrients within dust to the ocean and land surfaces (Adebiyi et al., 2023; Kok et al., 2017; Dansie et al., 2022). It is therefore important to understand and improve modelled dust size distributions. The discrepancy in size distribution could be due to over-active processes affecting the dust deposition, such as sedimentation, wet deposition, convective or turbulent mixing. It could also be a consequence of the dust not absorbing enough shortwave radiation (Colarco et al., 2014; Balkanski et al., 2021) and po-

tentially affecting heating and therefore dust lofting after emission or plume height during transport. Alternatively, this long range transport could be due to processes not considered in the model and not yet fully understood in practice, such as electric charging (van der Does et al., 2018; Toth III et al., 2020), asphericity (Huang et al., 2020, 2021; Saxby et al., 2018), turbulence (Denjean et al., 2016; Cornwell et al., 2021) and vertical mixing (Gasteiger et al., 2017). There is a need to better understand and observe dust size distributions at emission, and in the lowest layers of the atmosphere over source regions. Whilst model dust concentration and size distribution near sources could be improved by re-tuning the emissions scheme, this is unlikely to affect the evolution of size distribution with transport, where additional processes are necessary to retain coarser particles, and should be investigated in further research along with size-resolved dust emissions.

This study presents an in-depth analysis of the evolution of the vertically resolved dust size distribution from the Sahara to the Caribbean from aircraft observations and the Met Office Unified Model. We show that the model underestimates super-coarse particles over the Sahara compared to observations, a difference which is exacerbated by up to 5 orders of magnitude during trans-Atlantic transport. As the presence and relative fraction of coarse particles is important for, among other processes, the Earth's radiative budget and ice nucleation, it is imperative for the scientific community to obtain a better understanding of the physical processes which could be better understood and/or improved for models to improve simulations of super-coarse dust transport. The work presented here demonstrates the need for thorough analysis of processes affecting dust transport and deposition across the Atlantic in both observations and modelling, in order to fully constrain models and to accurately simulate dust size changes during long range transport, and the diverse impacts on weather, climate and socio-economics dependent on this.

*Data availability.* Aircraft data for the Fennec and AER-D campaigns are available at the Center for Environmental Data Archive at https://dx.doi.org/10.5285/1f4555d2589841a8a4bbbf1fe42f54c8 and http://catalogue.ceda.ac.uk/uuid/d7e02c75191a4515a28a208c8a069e70, respectively. Aircraft data for the SALTRACE campaigns are currently in the process having a doi created. Data from the model simulation are available at https://doi.org/10.5281/zenodo.10722717

*Author contributions.* NGR, CLR and NB designed the research. NGR carried out the analysis and wrote the paper. All authors discussed the methodology and results. BW, LMW, JG and MD provided the SALTRACE data. All authors read and commented on the paper.

*Competing interests.* The authors declare that they have no conflict of interest.

*Acknowledgements.* NGR would like to acknowledge the funding and support given by CASE funding from the UK Met Office. Further detailed analysis of the SALTRACE data forms part of LMWs PhD thesis. LMW, MD and BW acknowledge funding through the University of Vienna and the Gottfried-and-Vera Weiss Foundation and FWF in the context of the PlasticSphere project (AP3641721). NGR acknowledges the use of the MONSooN system, a collaborative facility supplied under the Joint Weather and Climate Research Programme, a strategic partnership between the Met Office and the Natural Environment Research Council. The MODIS data in this study were acquired as part of NASA's Earth Science Enterprise. The algorithms were developed by the MODIS Science Teams, and the data were processed by the MODIS Adaptive Processing System (MODAPS) and Goddard Distributed Archive Centre (DAAC); they are archived and distributed by the Goddard DAAC.

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
