# Peer review of "Long range transport of coarse mineral dust: an evaluation of the Met Office Unified Model against aircraft observations"

_EGUsphere, 2024_

## Author Comment (AC1)

**Response to Reviewers**

We thank the reviewers for their thoughtful and constructive comments. Below we respond to each point raised after briefly responding to an issue raised by both reviewers.

We would like to note to the reviewers and editor that there has been a small correction to some of the SALTRACE campaign data as a result of slight corrections to data processing methods. These changes have been applied to the figures and text in this manuscript and do not change the conclusions of the research. Two additional flights have been added to Table 4. The total dust mass profiles in Figure 2 are only minorly changed; the new profiles are not significantly discernible from the previous profiles and the MCA values have changed by only ~30 m altitude. There is now a slightly smaller proportion of size bin 6 particles which results in slight changes to Figure 3 and the associated values in Table 8. At Cape Verde, the size bin 6 percentage contribution is reduced by 10% and at the Caribbean, it is only reduced by 5%.

Both reviewers requested more detail regarding the dust emissions setup in the model. In order to provide a better description of the emissions process, we have added a substantial amount of information on the calculation of the emissions in the methods section. This includes the equations used to calculate the horizontal and vertical fluxes of dust, and further information on the interactive calculation of the dust size distribution.

Additionally, we now add a supplementary plot showing the emitted mass size distribution at the Sahara, as well as the mass size distribution in the first atmospheric layer, which shows the overly swift loss of the coarse particles immediately after emission.

Figure to be included in supplementary material

[Figure]

Figure S1: Model emitted dust mass (dashed blue line) and the mass size distribution (red solid line) in the lowest atmospheric level (0-36 m) at the Sahara (10W-25E, 18-29N).

**Reviewer Comment 1**

The paper presents data on observed dust particle size distribution vertical profiles from three airborne campaigns: Fennec, AER-D, and SALTRACE. The data present evidence for the existence of super-coarse and giant sized dust particles from their origins over the Sahara desert. The data from these different campaigns is synthesized to be evaluated in context of climate model simulations performed with the HadGEM3 model. The model is shown to have systematic biases with respect to how it apportions coarse and fine mode dust, emphasizing too much the fine mode dust at the expense of coarse particles. This is a result of model tuning (mentioned, not shown) that itself corrects a bias toward the model's excessive loss of coarse dust particles compared to observations. In short, the model is tuned to satellite AOD. A statistical analysis shows the positive correlation relationship between dust AOD and dust concentration is robust and that there is no strong evidence of a relationship between particle size and dust AOD (in other words, the dust particle size is similar regardless of loading).

The author's are writing on a timely topic and using the best available airborne observations. The modeling methodology needs further discussion and I suggest a major revision only because I'm going to ask them to do some additional calculations, but I don't think they should be too burdensome:

We thank the reviewer for their detailed comments. Responses to each specific point can be found below in blue text.

We have taken the opportunity to clarify some details on the model tuning in the manuscript at line 258: "The model dust emissions are tuned to improve agreement between the simulation and observations of AOD, near-surface concentrations and deposition rates"

Further detail is needed in the paper to describe aspects of the modeling. First, what is the initial size distribution for dust emission assumed? Then, what is the effective radius of the dust size bins that is used in the settling calculation? These could be added straightforwardly to Table 5. But further I suggest the analysis will be improved if a fundamental model bias is corrected at the start. Specifically, as Figure 4a shows the model doesn't even agree with the size distribution over the source region. I think this is consistent with Figure 3b in the Woodward et al. (2022) paper referenced in this paper. It does not seem as though the model would under any circumstances be capable of reproducing the observed size distributions because at the source the particles are biased toward smaller sizes. It is not clear to me from reading the Woodward paper how the emitted size distribution is calculated; the paper seems to suggest it is related to the horizontal dust flux at the surface, but I don't think that's right (or indeed what is shown).

As mentioned above, we have added additional information to the methods section about how the emitted size distribution is calculated. Full details are now given in the paper, including information on how the vertically emitted size distribution depends on the horizontal flux. The emitted size distribution is variable, dependent on wind speed, soil type, soil moisture. Table 5

now contains the representative diameter for each size bin which is used in calculating the emitted size distribution and settling velocity.

We have added a supplementary figure (see above) showing the emitted size distribution and the size distribution in the lowest atmospheric model layer. While the emissions are dominated by the larger size bins, the size distribution in the lowest model layer tails off in bins 5 and 6, being dominated by size bin 3. While we do not focus on evaluating emissions, Figure 1c in Kok et al. (2017) suggests that the shape of the model emitted size distribution seems reasonable. Therefore we might infer that it is the atmospheric processes which are responsible for the differences seen in our Figure 4, implying that the model *does* have a chance of replicating the correct size distribution.

My request then is that here the authors make a version of Figure 4 in which they rescale the modeled dust according to the initial dust particle size distribution observed in the Fennec measurements over the Sahara. At least near the surface. Applying those same scaling factors it should be possible to make a version of Figure 4 that looks in some way more like the observed dust, but still makes the point that the coarse mode particles fall out too quickly. It would demonstrate that the model itself has this deficiency and it is not simply a function of incorrect assumptions for the initial particle size distribution. The model would need to be retuned if such were to be used more routinely, but since in this paper everything is scaled and normalized to facilitate the comparisons it is just one more example of that. I recommend this would be clarifying for the literature on this topic.

We agree with the point that the model underestimates the PSD over the Sahara, therefore would be unlikely to represent the PSD accurately further downstream, and welcome the opportunity to clarify the changes during transport in the model. In order to demonstrate the additional discrepancy between model and observations during transport, over and away from the Sahara, we examine the relative changes between the model and observations as a function of longitude (or region). We plot the fractional differences between the observations and the model at each location. This figure (new Figure 5, see below) shows the increasing underestimation of the model compared to observations of all sizes during transport, but especially the coarse particles, where the fractional difference between model and observations for size bin 6 increases from around a factor of 10 over the Sahara to a factor of over one million over the Caribbean. We believe that this demonstrates the important changes during transport to the same effect as those suggested by the reviewer.

[Figure]

Additionally, we note that the observations are not made at the surface, and our campaign aggregated Fennec observations are upwards from 1000 m.

Line 37: Haven't yet identified why the simulation in models is "challenging." Suggest you remove this phrase here.

We have removed this term.

Line 55: See Nowottnick et al. 2010 for an example of this that was found to require improvement of wet scavenging processes.

Nowottnick, E., P. Colarco, R. Ferrare, G. Chen, S. Ismail, B. Anderson, and E. Browell (2010), Online simulations of mineral dust aerosol distributions: Comparisons to NAMMA observations and sensitivity to dust emission parameterization, J. Geophys. Res., 115, D03202, doi:10.1029/2009JD012692.

Following the reviewer's suggestion, we have added this citation to the introduction of the manuscript (Line 72).

Line 61 - 74: There is no mention here about how the processes are represented in the models, i.e., the numerics of the settling calculation, for example. See for example Ginoux (2003) who shows significant difference between simple upwind advection scheme for calculating particle vertical transport versus a higher order scheme (his Figure 4b). This has to matter to solving this problem. What kind of scheme is used in your model?

Ginoux, P., Effects of nonsphericity on mineral dust modeling, J. Geophys. Res., 108 (D2), 4052, doi:10.1029/2002JD002516, 2003.

We have added the following text to provide more information on the numerical schemes used. Line 251: " The impact of gravitational settling on the distribution of dust mass is calculated by computing the flux of dust out of a given layer and down to up to two model levels below (determined partly by the vertical spacing of the model levels), in proportion to the stokes velocity and the length of the timestep. The sensitivity of model results to the precise numerics have not been tested."

We have also added the following text to the conclusions. "There is the potential for future research into the sensitivity of coarser particle transport in the model to the numerical schemes which could provide additional valuable information to this research topic."

Line 125: You mention sizes up to 300 um here (and on line 142) but I don't see that in Table 3. What do you mean here? Do you have any quantitative information about such large sizes?

300 um refers to the maximum size in the observations, while we restrict the maximum size examined to match up to the maximum model size bin (63.2 um). The following has been added in the methods section to clarify the use of a limited size range as opposed to the total size range available from the observations. (Line 127) "Both the Fennec and AER-D campaigns measured particles up to 300 um diameter. In order to tailor our analysis to the model, only observations corresponding to the model size bins (up to 63.2 um diameter) are used in this study."

Table 3 was previously missing information on the CIP15 instrument; this has now been added.

Table 5: Could you provide some additional details of the model here? For example, how total vertical flux is partitioned into each size class (i.e., could your model potentially *ever* match the size distribution of observed dust shown in Figure 4), and also what the radius used for the settling velocity calculation is (and/or what is the settling velocity for each bin at, say, 1-2 km altitude).

Please see earlier responses concerning horizontal and vertical flux partitioning. We hope that the changes made in the model methods section discussed earlier satisfy the comments made here by the reviewer, as well as the discussion above of whether the model could potentially ever match the observations at the source. We have added a 'representative diameter' for each size bin used in the sedimentation calculations to Table 5. We would also like to note that the model can be tuned to better match observations at the Sahara, for example in Figure 3a of Woodward et al. (2022), however we are observing the model departing from the observations with distance from the source.

Line 266: I'm don't understand what you are saying differently in this paragraph than in the following one (line 274). Here you explicitly refer to spatial distribution, but the result is presented in terms of mean AOD differences. You say those are comparable between the campaign and long-term averages except for the Canaries. You make these same points in the next paragraph, although there you also say SALTRACE observed dustier than typical conditions. How do you assess comparable spatial patterns? And what is it is really you are seeing? Dustier than average conditions, or typical conditions? This could be clarified.

The paragraphs mentioned here have been edited and combined to read more clearly (Lines 290-302). The text has been edited to remove reference to spatial variability since we are examining temporal changes for our specific observational regions.

Line 294: The stated null hypothesis reads differently than what is written in caption 6. The statement in the caption reads more sensibly than what is written here. Please adjust. You conclude statistically significant differences in all cases, although the Canaries case shows no significant difference when M and H cases are compared; why is that?

This has been rewritten and clarified (Line 318). Unfortunately, we are not confident as to why the M and H cases at the Canaries are significantly more similar. Thus, we have changed our wording here to state that it is not 'all' cases which show statistically significant differences.

Line 305: You mention Figure 3 before Figure 2. Suggest you just reorder the figures.

This has been rectified in the text.

Lin 325: The appears to be an inconsistency here between the stated mean mass concentrations and the mean plotted on Figure 3. For example, the stated mean concentration over the Sahara is 347 ug m-3, but Figure 3a shows values in excess of 500 ug m-3 from the surface to 5 km. Similar for other values. What is going on? Is the model calculation consistent?

We thank the reviewer for pointing this mistake out. The values have been fixed in the text and plots.

**Reviewer Comment 2**

The paper presents relevant data from various observation campaigns on dust particle size distribution vertical profiles, which are synthesized and analyzed to verify and contrast how can be crucial for the radiative aspects of climate models. The information and analysis from the campaigns are significant for furthering our understanding of Saharan dust transport, and the comparison with the model is particularly interesting for identifying uncertainties and limitations.

The manuscript is well-written, well-organized, and the presented analysis is accurate. I would like to propose a few questions to the authors that I believe are pertinent to the work conducted, as well as some minor considerations to enhance the reader's understanding of the paper.

We thank the reviewer for their positive comments. Responses to specific points are found below in blue text.

Questions:

What type of emission data is input into the model, and in what format, spatial resolution, temporal resolution, etc.? One key reason for the model's discrepancies regarding size distribution could be improper emission inputs.

As discussed at the beginning of the author response, we have added significantly more information regarding the emissions in the model. Emissions are calculated interactively online in the model using model wind fields, bare soil fraction and soil moisture. Temporal and spatial resolution of emissions are identical to that of the model.

We also note (and expand upon in the response to reviewer 1) that although the model does not accurately represent the dust size distribution over the Sahara, we focus on the additional discrepancy which develops during trans-Atlantic transport, over and above emission differences.

Given that the measurement campaigns are conducted on specific days and hours, have the results of these specific campaigns been compared with the model outputs for those concrete days? Perhaps the unaveraged model output is less representative but could provide clues to identify some of its limitations. Along these lines, can CTM models, which can be used with hourly resolution, be compared similarly to the climate model?

We have aimed to clarify our reasoning for not being able to carry out specific testing as suggested by the reviewer. We have included the following details in the text for clarification. Line 266: "The model is free-running, but uses observed SSTs to simulate five June months, 2010-2014, which outputs vertically resolved daily mean dust mass mixing ratios for each size bin. The averaged five Junes provide a 'June climatology' which is used to compare with our campaign averages. As the model is free-running, it does not represent specific meteorology and dust events, and therefore we cannot compare the specific dates on which the measurements were taken."

What is the vertical resolution used? The number of layers is mentioned, but I think it's relevant to include this to understand how the model outputs have been averaged. Are these models sensitive to increasing vertical resolution to improve results?

The following has been added to clarify information regarding the vertical resolution of the model and the importance of vertical numerical diffusion in this setup. Line 219: "The finest vertical resolution is the lowest layer, with a depth (dZ) of 36 m. dZ increases with altitude so that at ~500 m altitude, dZ is 120 m, at ~2 km altitude, dZ is 226 m and at~ 5 km altitude, dZ is 373 m. The relatively high vertical resolution suggests that sensitivity to vertical numerical diffusion is unlikely to be important, though this may have a small effect (Zhuang et al., 2018)." Additionally, a statement clarifying the treatment fo the vertical data in the model has been added at Line 279 "The model data is not averaged in the vertical".

The discrepancy between the model and observations is a recurring issue in the manuscript; it might be beneficial to summarize this in one or two paragraphs towards the end, if the authors agree.

We provide a summary of the main model vs observation differences in the third paragraph of the conclusions. These have been made more comprehensive by adding a summary of the differences in size-resolved profiles between observations and model.

The final paragraph of the conclusions feels somewhat weak and lacks impact. Given the importance of this work, it would be advisable to strengthen this last paragraph.

We thank the reviewer for this kind comment and have strengthened our concluding paragraph (Lines 621-630).

Recommendations to facilitate understanding of the paper:

Do SABL and SAL mean the same thing?

We have added a clarification at Line 367 at the first use of SABL for clarification "...the Saharan atmospheric boundary layer (SABL)--a well-mixed, dry layer over the Sahara extending from the surface, often up to ~6 km over the Sahara (Cuesta et al., 2009) ....... the SAL--the dry, dusty air layer formed when the SABL rises isentropically over the Atlantic ocean's MBL (Carlson, 2016), residing between ~1-6 km"

In line 388, "too low" is mentioned; I recommend providing height values as a reference for what is considered "too low."

We have added additional detail regarding the findings of this citation at line 415: "O'Sullivan et al. (2020) previously found that an NWP GA6.1 configuration of the MetUM placed dust *0.5-2.5 km* too low in the atmosphere *when compared with observations*"

Figure 3: I understand that the scales are correct due to the significant difference in mass concentration between the observations and the model; nonetheless, it would be helpful to include a note or warning regarding the scales in the figure.

Statement added to figure caption to clarify the use of different scales for the mass concentration lines. "It should be noted that mass concentration scales (black line) differ between panels."

Figure 5: It might be useful to mark the start of the distributions in Figure (a), since (b) and (c) start at 0° and (a) starts at 10°E. Alternatively, align (b) and (c) with the scale of (a) for better visual consistency.

Figure 5 (now Figure 6) has been adjusted as suggested by the reviewer so that (a) starts at 0° for continuity between the subplots.

The values in Table 8 are mainly referenced as sums in the following paragraph; consider indicating this either in the paragraph or in the table itself.

From Line 428 onwards additional clarifications have been made in the text and table caption to make it clear which size bins are being discussed and where they we are referencing multiple bins in sum.